# Characterizing Out-of-Distribution Error via Optimal Transport

**Yuzhe Lu**[*1], **Yilong Qin**[*1], **Runtian Zhai**[1], **Andrew Shen**[1], **Ketong Chen**[1]
**Zhenlin Wang**[1], **Soheil Kolouri**[2], **Simon Stepputtis**[1], **Joseph Campbell**[1], **Katia Sycara**[1]
[1]Carnegie Mellon University   [2]Vanderbilt University
{yuzhelu, yilongq, rzhai, andrews, ketongc, zhenlinw}@cs.cmu.edu
soheil.kolouri@vanderbilt.edu, {ssteptut, jcampbell, sycara}@cs.cmu.edu

## Abstract

Out-of-distribution (OOD) data poses serious challenges in deployed machine learning models, so methods of predicting a model's performance on OOD data without labels are important for machine learning safety. While a number of methods have been proposed by prior work, they often underestimate the actual error, sometimes by a large margin, which greatly impacts their applicability to real tasks. In this work, we identify *pseudo-label shift*, or the difference between the predicted and true OOD label distributions, as a key indicator of this under-estimation. Based on this observation, we introduce a novel method for estimating model performance by leveraging optimal transport theory, Confidence Optimal Transport (COT), and show that it provably provides more robust error estimates in the presence of pseudo label shift. Additionally, we introduce an empirically-motivated variant of COT, Confidence Optimal Transport with Thresholding (COTT), which applies thresholding to the individual transport costs and further improves the accuracy of COT's error estimates. We evaluate COT and COTT on a variety of standard benchmarks that induce various types of distribution shift – synthetic, novel sub-population, and natural – and show that our approaches significantly outperform existing state-of-the-art methods with up to 3x lower prediction errors. Our code can be found at https://github.com/luyuzhe111/COT.

## 1   Introduction

Machine Learning methods are largely based on the assumption that test samples are drawn from the same distribution as training samples, providing a basis for generalization. However, this i.i.d. assumption is often violated in real-world applications where test samples are found to be out-of-distribution (OOD) – sampled from a different distribution than during training. This may result in a significant negative impact on model performance [15, 34, 25, 38]. A common practice for alleviating this issue is to regularly gauge the model's performance on a set of labeled data from the current *target* data distribution, and update the model if necessary. When labeled data is not available, however, one needs to predict the model's performance on the target distribution with unlabeled data, a task known as *OOD performance prediction*.

Performance prediction on unlabeled data has previously been shown to be impossible without imposing additional constraints over the unknown target distribution [7, 11, 5, 26], due to the fact that target samples may take any label. Thus, the feasibility of this task is dependent on what assumptions we make regarding the shift between the train and target distributions. Prior works often make the assumption that the conditional density $P(y|x)$ remains fixed in the presence of covariate shift [37].

---

[*]These authors contributed equally to this work.

37th Conference on Neural Information Processing Systems (NeurIPS 2023).

However, this tells us little when $x$ falls outside the support of the train distribution. Despite the theoretical difficulty, prior works have proposed a number of heuristic methods to estimate the performance of a model based on unlabeled target samples. For instance, Average Confidence (AC) [20, 16] estimates error based on the average maximum softmax score for target samples, assuming the model has been calibrated for the train distribution. This method was further improved with the addition of a learned threshold [13], for which the error is predicted as the fraction of samples with confidence falling below it. Other approaches estimate model performance based on a disagreement score computed between the predictions of two models trained over the same dataset [22, 3]. Some works have found that applying a transformation over target samples and estimating the effect of the transformation leads to a reliable prediction of model performance in vision tasks [9, 10]. However, many of these prior methods have been shown to underestimate the model's error when it is *miscalibrated* in the target distribution [13, 22]; that is, the predicted softmax scores differ from the true class likelihoods. We empirically observe that this miscalibration is strongly positively correlated with *pseudo-label shift*, which is the difference between the predicted target label distribution $P_T(\tilde{y})$ and the true label distribution $P_T(y)$. Thus, we treat pseudo-label shift as a key indicator of error underestimation.

In this work, we propose an approach which provides robust error estimates in the presence of pseudo-label shift. Our approach, Confidence Optimal Transport (COT), leverages the optimal transport framework to predict the error of a model as the Wasserstein distance between the predicted target class probabilities and the true source label distribution. We theoretically derive lower bounds for COT's predicted error. This results in a more provably conservative error prediction than AC, which is crucial for safety in many real-world machine learning applications. In addition, we introduce a variant of COT, Confidence Optimal Transport with Thresholding (COTT), which introduces a learned threshold over the optimal transportation costs in line with prior work [13] and empirically improves upon COT's performance. In this work, we mainly target covariate shifts and compare our proposed methods to existing state-of-the-art approaches in extensive empirical experiments over eleven datasets exhibiting distribution shift from both vision and language domains. These distribution shifts include: visual corruptions (e.g., blurred image data), novel subpopulation shifts (e.g., novel appearances of a category), and natural shifts in the wild (e.g., different stain colors for pathology images), all of which are frequently experienced in the real world. We find that COT and COTT consistently avoid the significant error underestimations suffered by previous methods. In particular, COTT achieves significantly lower prediction errors than existing methods for most models and datasets (up to 3x better), establishing new state-of-the-art results.

## 2 Preliminaries and Motivation

In this section, we introduce the problem setup of OOD error prediction and show how a popular baseline, Average Confidence (AC), can be understood as the Wasserstein Distance (WD) between a reference label distribution and the softmax output distribution from an Optimal Transport (OT) perspective. We then demonstrate why the reference label distribution AC uses is problematic by explaining pseudo-label shift and its correlation with miscalibration. After establishing the relation and utility of OT to OOD error prediction, we formally introduce our proposed methods in Sec. 3.

### 2.1 OOD Performance Prediction

In this work, we address the OOD problem in the domain of classification tasks. Let $\mathcal{X} \subseteq \mathbb{R}^d$ be the input space of size $d$, and $\mathcal{Y} = \{1, \cdots, k\}$ be the label space, where $k$ is the number of classes. Let the source distribution over $\mathcal{X} \times \mathcal{Y}$ be $P_S(x, y)$ and the target distribution be $P_T(x, y)$. A classifier $\vec{f} : \mathcal{X} \to \Delta^{k-1}$ maps an input to a *confidence vector* (i.e. the output of the softmax layer), where $\Delta^{k-1} = \{(z_1, \cdots, z_k) : z_1 + \cdots + z_k = 1, z_i \geq 0\}$ is the $k$-dimensional unit simplex. Let the training samples be $\{(x_{\text{train}}^{(i)}, y_{\text{train}}^{(i)})\} \sim P_S(x, y)$, and the validation samples be $\{(x_{\text{val}}^{(i)}, y_{\text{val}}^{(i)})\} \sim P_S(x, y)$. The validation set is used in some previous methods and will also be used in COT and COTT to perform calibration [17].

The OOD performance prediction problem is formally stated as follows: Given an unlabeled test set $\{x_T^{(i)}\} \sim P_T(x)$, and a classifier $\vec{f}$ trained over the training samples, predict the accuracy of

$\vec{f}$ over the test set $\alpha = \frac{1}{n} \sum_{i=1}^{n} \mathbb{1}[y_T^{(i)} = \arg\max_j \vec{f}_j(x_T^{(i)})]$, where $y_T^{(i)}$ is the ground truth label. Equivalently, we can also predict the error $\epsilon = 1 - \alpha$ with an estimate $\hat{\epsilon}$.

In this work, we are further investigating *the distribution of confidence vectors*. While a confidence vector $\vec{f}(x)$ itself is a distribution of labels in the $k$-dimensional simplex $\Delta^{k-1}$, the distribution of confidence vectors $\vec{f}_{\#}P(\vec{c})$ is a distribution of distributions, where $\vec{c} \in \Delta^{k-1}$. $\vec{f}_{\#}P(\vec{c})$ is defined to be the *pushforward* of a covariate distribution $P(x)$ using $\vec{f}$: $\vec{f}_{\#}P(\vec{c}) = P(\vec{f}^{-1}(\vec{c}))$. Consider the following example: Suppose we have a uniform covariate distribution $P(x)$ on $x \in \{A, B, C\}$ and $\vec{f}$ maps $A, B$ to $[0.5, 0.5]^\top$ and $C$ to $[0.7, 0.3]^\top$. Then, $f_{\#}P(\vec{c})$ will assign $\frac{2}{3}$ mass to $[0.5, 0.5]^\top$ and $\frac{1}{3}$ mass to $[0.7, 0.3]^\top$. To facilitate easy comparison between confidence vectors and labels, we denote $\vec{y} \in \{0, 1\}^k \cap \Delta^{k-1}$ to be the one-hot representation of $y \in \mathcal{Y}$, where the only non-zero element in $\vec{y}$ is the $y$-th element, i.e. $\vec{y}_j = \mathbb{1}[y = j]$. We denote *the distribution of one-hot labels* as $P(\vec{y})$. For a covariate distribution $P(x)$, we will refer to the distribution of predicted labels from classifier $\vec{f}$ as the *pseudo-label distribution* $P_{\text{pseudo}}(\vec{y})$. We reuse the notation to denote the probability mass of $\vec{y}$ also as which is given by the mass of the inputs that give this prediction, i.e. $P_{\text{pseudo}}(\vec{y}) = P(\{x \in \mathcal{X} | \arg\max_j \vec{f}_j(x) = y\})$.

## 2.2  Wasserstein Distance and Optimal Transport

In recent years, optimal transport theory has found numerous applications in the field of machine learning [2, 4, 1, 27, 29]. Optimal transport aims to move one distribution of mass to another as efficiently as possible under a given cost function. In the Kantorovich formulation of optimal transport, we are given two distributions $\mu(x)$ over $\mathcal{X}$ and $\nu(y)$ over $\mathcal{Y}$ and a cost function $c(x, y)$ that tells us the cost of transporting from location $x$ to location $y$. Here, we aim to find a transport plan $\pi(x, y)$ that minimizes the total transport cost. The transport plan is a joint distribution with marginal $\pi(\cdot, \mathcal{Y}) = \mu(\cdot)$ and $\pi(\mathcal{X}, \cdot) = \nu(\cdot)$. The conditional distribution $\frac{\pi(x,y)}{\mu(x)}$ of the transport plan informs us how much mass is moved from $x$ to $y$. Let $\Pi(\mu, \nu)$ be the set of all transport plans. More formally, the Wasserstein Distance is defined as:

$$W(\mu, \nu) = \inf_{\pi \in \Pi(\mu, \nu)} \int_{\mathcal{X} \times \mathcal{Y}} c(x, y) d\pi(x, y)$$

The Wasserstein distance satisfies the definition of a metric (non-negativity, symmetry, and sub-additivity), inducing a metric space over a space of probability distributions. Unlike other metrics, such as total variation, the Wasserstein metric induces a weaker topology and provides a robust framework for comparing probability distributions that respect the underlying space geometry [2].

For discrete distributions $\mu, \nu$ such as the empirical distributions, the Wasserstein distance simplifies to the following linear programming problem:

$$W(\mu, \nu) = \min_{\mathbf{P} \in \Pi(\mu, \nu)} \langle \mathbf{P}, \mathbf{C} \rangle = \min_{\mathbf{P} \in \Pi(\mu, \nu)} \sum_{i,j} \mathbf{P}_{ij} \mathbf{C}_{ij}$$

where $\mathbf{C}, \mathbf{P} \in \mathbb{R}^{m \times n}$ are the cost matrix and the plan matrix respectively and $\mathbf{C}_{ij}$ is the transport cost from sample $i$ to sample $j$. When $n = m$, the optimal transport problem reduces to the optimal matching problem (Proposition 2.1 in Peyré et al. [32]), i.e. the optimal transport plan $\mathbf{P}^* = \arg\min_{\mathbf{P} \in \Pi(\mu, \nu)} \langle \mathbf{P}, \mathbf{C} \rangle$ is a permutation matrix. Not only does this constraint enable efficient algorithms like the Hungarian algorithm, but it will also help draw the connection between pseudo-label shift and target error that is central to our Confidence Optimal Transport method.

## 2.3  Average Confidence as Wasserstein Distance

We use Average Confidence, a popular OOD performance prediction method, as the starting point of our analysis. Average Confidence with Max Confidence (AC-MC) estimates the target accuracy by taking the empirical mean of maximum confidence of the classifier over all target samples $x^{(i)} \sim P_T(x)$, i.e. $\frac{1}{n} \sum_{i=1}^{n} \max_j \vec{f}_j(x^{(i)})$. Its corresponding target error estimate is therefore $\hat{\epsilon}_{\text{AC}} = 1 - \frac{1}{n} \sum_{i=1}^{n} \max_j \vec{f}_j(x^{(i)})$. By definition, $|\epsilon - \hat{\epsilon}_{\text{AC}}|$ measures the miscalibration of the model, i.e. how far away the model's confidence is from its actual accuracy. In the following proposition, we connect the AC-MC estimates with distances in the Wasserstein Metric Space:

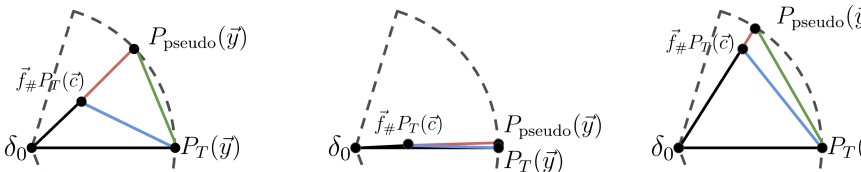

Figure 1: The AC-COT Triangle in the Wasserstein Space $(\mathcal{P}(\mathbb{R}^k), W_\infty)$, where $\delta_0$ (Dirac delta over the zero vector) represents the origin. Red line: AC-MC error estimate. Blue line: (our) COT error estimate (assuming $P_T(\vec{y}) \approx P_S(\vec{y})$). Green line: Pseudo-label shift. **Left:** AC-MC predicts the target error as the distance between the *distribution of confidence vectors* and its projection on the unit sphere, the smallest among all label distributions. This makes AC-MC prone to *underestimating* the target error. **Middle:** Mild pseudo-label shift. **Right:** Severe pseudo-label shift.

**Proposition 1** ($\hat{\epsilon}_{AC}$-$W_\infty$ Equivalence). *Let* $(\mathcal{P}(\mathbb{R}^k), W_\infty)$ *be the metric space of all distributions over* $\mathbb{R}^k$, *where* $W_\infty$ *is the Wasserstein distance with* $c(x, y) = \|x - y\|_\infty$. *Then, the estimated error of AC-MC is given by* $\hat{\epsilon}_{AC} = W_\infty(\vec{f}_\# P(\vec{c}), P_{pseudo}(\vec{y}))$.

We defer all proofs to the supplementary material. Here, we appeal to pictorial intuition in Figure 1, where $\delta_0$ (Dirac delta over the zero vector) represents the origin. All one-hot label distributions, including $P_T(\vec{y})$ and $P_{\mathrm{pseudo}}(\vec{y})$, are represented as points on the (dashed) unit sphere around $\delta_0$. A distribution of confidence vectors $\vec{f}_\# P(\vec{c})$ is simply a point within the unit ball around $\delta_0$. The AC-MC accuracy estimate measures how far the point is to the origin $\delta_0$. Additionally, we can project the point to the unit sphere, resulting in the projection point $P_{\mathrm{pseudo}}(\vec{y})$, which is the pseudo-label distribution. The AC-MC error estimate measures *the length of the projection*.

**Corollary 1** ($P_{\mathrm{pseudo}}(\vec{y})$ is closest to $\vec{f}_\# P(\vec{c})$). *Let* $P'(\vec{y}) \in \mathcal{P}(\{0, 1\}^k \cap \Delta^{k-1})$ *be a one-hot label distribution. Then* $W_\infty(\vec{f}_\# P(\vec{c}), P'(\vec{y})) \geq W_\infty(\vec{f}_\# P(\vec{c}), P_{pseudo}(\vec{y}))$.

This suggests that AC-MC, among all possible reference one-hot label distributions, selects the closest one and reports the distance to be the predicted error. Thus, we can see that AC-MC is a very optimistic prediction strategy, so it is not surprising that AC-MC is widely reported to be over-estimating the performance on real tasks [13, 16, 22, 3].

## 2.4 Pseudo-Label Shift and its Correlation with Miscalibration

The pseudo-label shift is defined as $W_\infty(P_{\mathrm{pseudo}}(\vec{y}), P_T(\vec{y}))$, i.e. the distance from the pseudo label distribution $P_{\mathrm{pseudo}}(\vec{y})$ to the ground truth target label distribution $P_T(\vec{y})$, which is also the length of the green line segment in Figure 1. An important property is $W_\infty(P_{\mathrm{pseudo}}(\vec{y}), P_T(\vec{y})) \leq \epsilon \leq 1$, i.e. the pseudo-label shift is a lower bound of the true target error of the model (as true target error can be interpreted as the average transport cost of a suboptimal matching). We can clearly see how a large pseudo-label shift could potentially destroy the prediction of AC-MC from Figure 1 (right), where the red line segment is much shorter than the green one which should be a lower bound of the true error. This lower bound, however, can be loose when it is small, as shown in Figure 1 (middle).

Most existing prediction methods heavily rely on the model being well-calibrated, as pointed out by Jiang et al. [22], Garg et al. [13], yet the underlying difficulty is a lack of precise understanding of *when* and *by how much* neural networks become miscalibrated. This is evident in the large-scale empirical study conducted in [30], showing that the variance of the error becomes much larger among the different types of shift studied, despite some positive correlation between the expected calibration error and the strength of the distribution shift. In this section, we empirically show that there is a strong positive correlation between the pseudo-label shift and $|\epsilon - \hat{\epsilon}_{AC}|$, which we define as the model's miscalibration. Because of this correlation, we consider pseudo-label shift as the key signal to why existing methods have undesirable performance.

We evaluate the pseudo label shift and the prediction error of AC-MC $|\epsilon - \hat{\epsilon}_{AC}|$ on CIFAR-10 and CIFAR-100 under different distribution shifts, and plot the results in Figure 2 (left). The plots demonstrate a strong positive correlation between these two quantities, which not only means that the performance of AC-MC worsens as the pseudo-label shift gets larger, but also implies that the performance of existing methods [16, 13, 22] that depend on model calibration will drop.

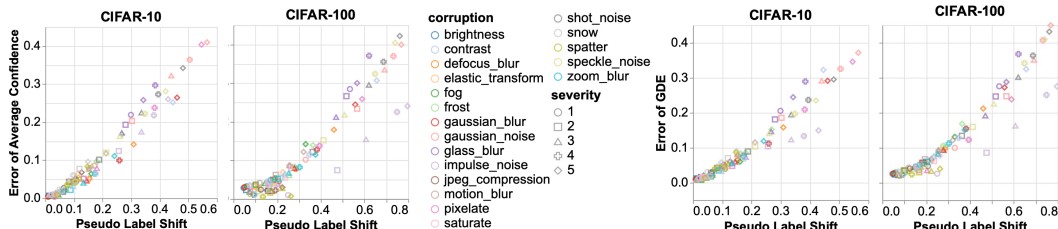

Figure 2: **Left:** Absolute difference between the model's Average Confidence and its true error under different levels of pseudo-label shift. This difference measures the degree of miscalibration. We see a strong correlation between the miscalibration and pseudo-label shift on the common corruption benchmarks (CIFAR-10-C, CIFAR-100-C). **Right:** Absolute difference between GDE error estimate and true error under different levels of pseudo-label shift. The strong correlation is also observed.

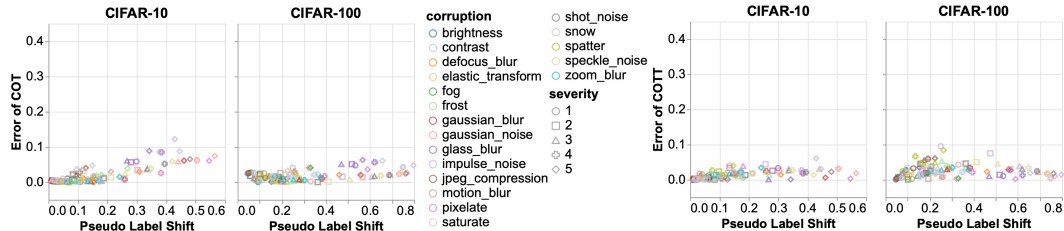

Figure 3: We further compare the sensitivity of COT and COTT's estimation error to the degree of pseudo-label shift. Compared to Figure 2, we can clearly see that the correlation between the prediction error and pseudo-label shift weakens significantly. Moreover, COTT is even more robust to pseudo-label shift compared to COT.

Our exploration of miscalibration and pseudo-label shift reveals a previously unexplored tradeoff: When the pseudo-label shift is small, prior work has given us some reassurance to trust the calibration of neural networks, which motivated methods such as AC [20] and ATC [13]. More critically, as miscalibration worsens, the pseudo-label shift becomes a tighter low bound of the error and thus a more trustworthy estimate of the error.

Motivated by this observation, we leverage the information of the pseudo-label shift to improve the performance of confidence-based prediction methods for miscalibrated models. The problem, as mentioned earlier, is that without any information about the target label distribution, it is theoretically impossible to estimate the pseudo-label shift when the test set contains data from unseen domains. Thus, the only option is to make assumptions on the target label distribution.

In our proposed method, we make a natural assumption: the target label distribution is close to the source label distribution. This assumption aligns with most natural shifts that can be observed in real-world datasets. This extra assumption allows us to develop COT and COTT, which perform much better than existing methods in most cases (See Table 1), especially when the pseudo-label shift is large. Given this assumption, Figure 3 (Left), shows the prediction error of COT versus the pseudo-label shift. We can see that equipped with the extra information, COT is able to maintain a low prediction error even under very large pseudo-label shift, and the correlation between COT performance and the pseudo-label shift is weak. Since the pseudo-label shift is strongly correlated with miscalibration, this implies that COT is much more robust to miscalibration than existing miscalibration-sensitive prediction methods.

## 3   Methods

In this section, we formally introduce our proposed method – Confidence Optimal Transport – and propose an additional variation, COTT, that utilizes a threshold over the optimal transportation costs between two distributions, instead of taking a simple average.

## 3.1 Confidence Optimal Transport

Let $\hat{P}_S(\vec{y})$ denote the empirical source label distribution. The predicted error of COT, which is the length of the blue line segment (assuming $P_T(\vec{y}) = P_S(\vec{y})$) in Figure 1, is given by

$$\hat{\epsilon}_{\text{COT}} = W_\infty(\vec{f}_\# \hat{P}_T(\vec{c}), \hat{P}_S(\vec{y})).$$

By Corollary 1, $\hat{\epsilon}_{\text{COT}} \geq \hat{\epsilon}_{\text{AC}}$, which means that COT provably predicts a larger error than AC-MC, which empirically tends to produce more accurate predictions as we find overestimation of the error far less common than underestimation. As mentioned earlier, in the case where the pseudo-label shift is large, such as in Figure 1 (Right), AC-MC can have arbitrarily large prediction error, while COT always has the following guarantee:

**Proposition 2** (Calibration independent lower bound of COT). *Under the assumption that $P_T(\vec{y}) = P_S(\vec{y})$, we always have $\hat{\epsilon}_{COT} \geq 0.5 W_\infty(P_{pseudo}(\vec{y}), P_T(\vec{y}))$.*

Thus, we can see that COT is by nature different from existing methods because it is a *miscalibration-robust method*. Its success is dependent on the difference between $P_T(\vec{y})$ and $P_S(\vec{y})$, which is relatively small in most real-world scenarios, while the dependency on calibration of existing methods can not always be controlled. The geometric explanation of this is that COT measures a fundamentally different length in the Wasserstein Space $(\mathcal{P}(\mathbb{R}^k), W_\infty)$, allowing for the above guarantee which does not exist in previous methods. Moreover, as we will empirically demonstrate in the next section, large pseudo-label shift is prevalent in real models and datasets. Consequently, COT performs much better than miscalibration-sensitive methods in most cases.

## 3.2 Thresholding as a Robust Transport Cost Stastistic

Computationally, COT (or Wasserstein Distance in general) is implemented as a two-step process: 1) calculating individual transport costs for all samples; 2) returning the mean across all transport costs as the error estimate. While we have seen that COT has some protection against miscalibration, it is not completely immune. Another outstanding issue lies in the second step - computing the mean of all transport costs. In statistics, it is well known that the mean is less robust to outliers than the median [21]. In the supplemental material, we show that the empirical transport cost distribution is also heavy-tailed and therefore COT is susceptible to outlier costs. A large fluctuation in the outliers impacts the mean much more than it does to the median. Therefore, a more robust statistic is desired to protect COT against these outliers.

To search for such a robust statistic that is suitable for our purpose, we turn to Average Thresholded Confidence (ATC) [13] for inspiration. ATC tremendously improves the performance of AC precisely by offering such protection against the overconfident outlier predictions (shown in supp material) by returning the percentile above the threshold rather than the mean. In a similar vein, we safeguard COT against outlier costs by introducing COT with Thresholding, COTT.

Specifically, to compute the threshold $t \in [0, 1]$, we first sample a validation set $\{(x_{\text{val}}^{(i)}, y_{\text{val}}^{(i)})\}_{i=1}^n \sim P_S(x, y)$. Let $\hat{P}_{\text{val}}$ denote the empirical validation distribution. We compute the optimal transport plan $\mathbf{P}_{\text{val}}^* = \arg\min_{\mathbf{P} \in \Pi(\vec{f}_\# \hat{P}_{\text{val}}(\vec{c}), \hat{P}_S(\vec{y}))} \langle \mathbf{P}, \mathbf{C} \rangle$, where $\mathbf{C}$ is the cost matrix determined by the L-infinity distance. Note that $\mathbf{P}_{\text{val}}^* \in \{0, 1\}^{n \times n}$ is a permutation matrix due to the equal number of confidence vectors and one-hot labels (Proposition 2.1 in [32]). Then the threshold $t$ is set such that the validation of error the classifier equals the fraction of samples with transport cost higher than $t$, i.e.

$$\epsilon_{\text{val}} = \frac{1}{n} |\{\mathbf{C}_{ij} \geq t | \mathbf{P}_{\text{val}ij}^* = 1\}|$$

With the threshold $t$ learned from the validation set, we can compute the COTT target error estimate. Let $\hat{P}_S(\vec{y})$ denote the empirical source label distribution and $\vec{f}_\# \hat{P}_T(\vec{c})$ denote the empirical distribution of confidence vectors of samples from the target distribution $P_T(x)$. We compute the optimal transport plan $\mathbf{P}^* = \arg\min_{\mathbf{P} \in \Pi(\vec{f}_\# \hat{P}_T(\vec{c}), \hat{P}_S(\vec{y}))} \langle \mathbf{P}, \mathbf{C} \rangle$. Our COTT estimate is then defined as

$$\hat{\epsilon}_{\text{COTT}} = \frac{1}{n} |\{\mathbf{C}_{ij} \geq t | \mathbf{P}^*_{ij} = 1\}|$$

**Note on Implementation:** While solving the linear program of optimal transport for COT and COTT is rather straightforward with existing solvers such as POT [12], its time complexity $\mathcal{O}(n^3)$ prohibits large-scale inputs. We bypass this problem by breaking the test dataset into small batches with a maximum size of 10,000. Concretely, given $n$ test samples, we sample with replacement for $\lceil n/10,000 \rceil$ batches and compute an error estimate for each batch. We get the final error estimation of COT and COTT by taking an average of batch estimates.

# 4 Experiments

In this section, we empirically compare COT and COTT with existing methods on various benchmark datasets. For all experiments, we trained the model on in-distribution data and froze the model after convergence. To predict the model's performance on the target domain, we only used unlabeled data from the target domain. When we have a test set size greater than 10,000, we show the results of utilizing the batched version of COT and COTT detailed in Section 3.2. For context, solving the OT problem of size 10,000 only takes around 10s, thus adding only negligible computation overhead.

## 4.1 Datasets and Nature of Shift

In our comprehensive evaluation, we consider more than 10 benchmark datasets across multiple modalities, including vision and language, with a variety of distribution shifts. Each type of shift is introduced in the following paragraphs:

**Synthetic Shift:** First, we consider distribution shifts caused by common visual corruptions, such as brightness, defocusing, and blurriness, which are common in real-world settings. We used the corrupted versions of CIFAR10, CIFAR100, and ImageNet proposed in [19], which includes 19 types of common visual corruptions across 5 levels of severity.

**Novel Subpopulation Shift:** Next, we consider novel subpopulation shifts, where the subpopulations in the train and test sets differ. For example, models might have only observed golden retrievers for the dog class but not huskies. In our experiments, we used the BREEDS benchmark [36], which leveraged the ImageNet[8] class hierarchy to create 4 datasets, Living-17, Nonliving-26, Entity13, and Entity-30. With this benchmark, we aim to understand if error estimation methods could capture challenges in an increasingly diverse world.

**Natural Shift:** Finally, we consider non-simulated shifts in which the distribution shifts are induced through differences in the data collection process, such as ImageNet-V2 and CIFAR10-V2 proposed in [34]. We also include ImageNet-Sketch [40], which consists of sketched images of the original ImageNet classes. Additionally, we consider distribution shifts faced in the wild, such as ones curated in the WILDS benchmark [24]. We consider four WILDS datasets, two for language tasks (Amazon-WILDS, CivilComments-WILDS) and two for vision tasks (Camelyon17, RxRx1). These datasets reflect various discrepancies between data sources happening in the real world. For example, hospitals might use different stain colors for pathological images. Different groups might follow distinct standards assigning star ratings for their reviews.

## 4.2 Architectures and Evaluations

For vision tasks, We trained ResNet18 (CIFAR10, CIFAR100) and ResNet50 [18] (ImageNet, Living17, Nonliving26, Entity13, Entity30, Camelyon17-WILDS, RxRx1-WILDS); for language tasks (Amazon-WILDS, CivilComments-WILDS), we fine-tuned DistilBERT-base-uncased [35]. We followed training setups from previous works [13] and provided the full details in the supplemental material. After training, we calibrated models using Temperature Scaling (TS) [17] on the in-distribution validation data, effectively adjusting the output probabilities of the neural network to match the actual correctness likelihood. This approach has previously demonstrated that TS consistently improves error estimation performance for all methods [13]. To evaluate different methods, we utilized the mean absolute difference between their predicted errors and the true errors, which are obtained using ground truth labels. We refer to this metric as Mean Absolute Error (MAE).

## 4.3 Baselines

We consider an array of baselines to compare against our methods: COT and COTT.

Table 1: Mean Absolute Error (MAE) between the estimated error and ground truth error to compare different methods. The "shift" column denotes the nature of distribution shifts for each dataset. For vision datasets, we reported results for ResNet18 and ResNet50; for language datasets, we reported results for DistilBERT-base-uncased. The results are averaged over 3 random seeds. We highlight the best-performing method. We defer the full table with std to the supplemental material.

| Dataset | Shift | Baselines | | | | | | Ours | |
|---|---|---|---|---|---|---|---|---|---|
| | | AC | DoC | IM | GDE | ATC-MC | ATC-NE | COT | COTT |
| CIFAR10 | Natural | 5.97 | 5.38 | 5.87 | 5.9 | 3.38 | **3.15** | 5.41 | 3.33 |
| | Synthetic | 9.1 | 8.53 | 9.26 | 8.84 | 4.2 | 3.37 | 2.17 | **1.7** |
| CIFAR100 | Synthetic | 10.83 | 8.76 | 12.07 | 11.36 | 6.8 | 6.63 | **2.09** | 2.59 |
| ImageNet | Natural | 8.5 | 7.43 | 8.62 | 5.62 | 3.57 | 2.6 | 3.88 | **2.41** |
| | Synthetic | 10.34 | 9.28 | 12.87 | 6.54 | 1.59 | 3.41 | 3.24 | **1.42** |
| Entity13 | Same | 19.63 | 19.2 | 17.5 | 15.37 | 8.09 | 7.23 | 8.47 | **2.61** |
| | Novel | 29.61 | 29.18 | 27.22 | 24.48 | 14.54 | 9.49 | 15.9 | **5.46** |
| Entity30 | Same | 16.97 | 16.21 | 13.56 | 13.98 | 8.19 | 9.08 | 5.9 | **2.46** |
| | Novel | 27.57 | 26.81 | 23.96 | 23.4 | 13.46 | 8.57 | 15.11 | **5.94** |
| Living17 | Same | 14.84 | 14.67 | 11.22 | 9.94 | 4.88 | 5.43 | 6.25 | **2.94** |
| | Novel | 29.61 | 29.18 | 27.22 | 24.48 | 14.54 | 9.49 | 15.9 | **5.53** |
| Nonliving26 | Same | 19.25 | 18.43 | 16.6 | 12.77 | 11.18 | 9.69 | 7.06 | **3.34** |
| | Novel | 31.37 | 30.54 | 28.79 | 23.37 | 19.93 | 16.56 | 17.8 | **10.46** |
| Camelyon17-WILDS | Natural | 9.44 | 9.44 | 10.24 | **5.19** | 7.73 | 7.73 | 7.27 | 5.71 |
| RxRx1-WILDS | Natural | 5.21 | 8.44 | 8.09 | 7.48 | 6.53 | 6.86 | **3.25** | 5.82 |
| Amazon-WILDS | Natural | 2.62 | 2.35 | 2.34 | 17.04 | 1.63 | **1.54** | 2.43 | 2.01 |
| CivilCom.-WILDS | Natural | 1.54 | 0.96 | **0.86** | 8.7 | 2.3 | 2.3 | 1.23 | 4.68 |

*Average Confidence (AC)* estimates target error by taking the average of one minus the maximum softmax confidence of target data. $\hat{\epsilon}_{AC} = \mathbb{E}_{x \sim \hat{P}_T}[1 - \max_{j \in \mathcal{Y}} \vec{f}_j(x)]$.

*Difference of Confidence (DoC a.k.a. DOC-Feat)* [16] estimates target error through the difference between the confidence of source data and the confidence of target data. $\hat{\epsilon}_{DoC} = \mathbb{E}_{x \sim \hat{P}_S}[\mathbb{1}[\arg \max_{j \in \mathcal{Y}} \vec{f}_j(x) \neq y]] + \mathbb{E}_{x \sim \hat{P}_T}[1 - \max_{j \in \mathcal{Y}} \vec{f}_j(x)] - \mathbb{E}_{x \sim \hat{P}_S}[1 - \max_{j \in \mathcal{Y}} \vec{f}_j(x)]$.

*Importance Re-weighting (IM)* estimates target error as a re-weighted source error. The weights are calculated as the ratio between the number of data points in each bin in target data and source data. This is equivalent to [6] using one slice based on the underlying classifier confidence.

*Generalized Disagreement Equality (GDE)* [22] estimates target error as the disagreement ratio of predictions on the target data using two independently trained models $\vec{f}(x)$ and $\vec{f'}(x)$. $\hat{\epsilon}_{GDE} = \mathbb{E}_{x \sim \hat{P}_T}[\mathbb{1}[\arg \max_{j \in \mathcal{Y}} \vec{f}_j(x) \neq \arg \max_{j \in \mathcal{Y}} \vec{f'}_j(x)]]$.

*Average Thresholded Confidence (ATC)* [13] first identifies a threshold $t$ such that the fraction of source data points that have scores below the threshold matches the source error on in-distribution validation data. Target error is estimated as the expected number of target data points that fall below the identified threshold. $\hat{\epsilon}_{ATC}(s) = \mathbb{E}_{x \sim \hat{P}_T}[\mathbb{1}[s(\vec{f}(x)) < t]]$, where $s$ is the score function mapping the softmax vector to a scalar. Two different score functions are used, Maximum Confidence (ATC-MC) and Negative Entropy (ATC-NE).

In addition, we also compare our method to *ProjNorm* [41]. ProjNorm cannot provide a direct estimate, instead, the authors demonstrated their metric has the strongest linear correlation to true target error compared to existing baselines. In this case, we followed their setup and performed a correlation analysis to draw a direct comparison. We included results in the supplemental material and showed that our method consistently outperforms ProjNorm.

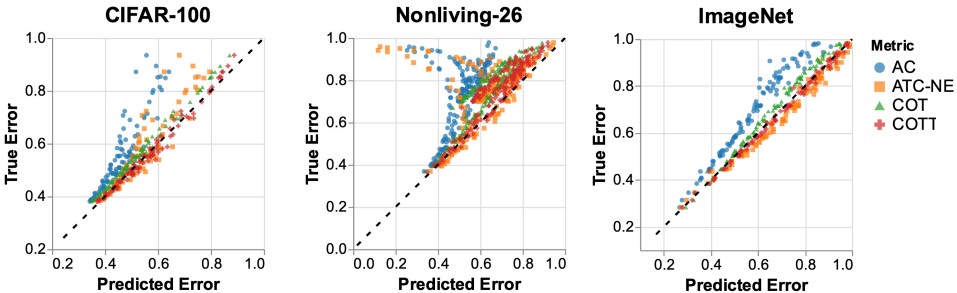

Figure 4: Qualitative results for AC, ATC, COT, and COTT, comparing error estimates vs. ground truth target error. Accurate estimates should be close to $y = x$ (dashed black line). Notably, COT and COTT remain accurate even when the shifts are large. By contrast, AC and ATC often severely underestimate the error, which is particularly evident in the Nonliving-26 dataset.

## 4.4 Results

In Table 1, we report the MAE results grouped by datasets and nature of shifts. Across all benchmarks, we observe that COT always obtains lower estimation error than AC, supporting our theoretical analysis that COT additionally leverages pseudo-label shift to fight miscalibration. On synthetic shift benchmarks (CIFAR10-Synthetic, CIFAR100-Synthetic, ImageNet-Synthetic BREEDS-same), COT is $2 - 4\times$ better than AC, drastically reducing the estimation error. On novel subpopulation shift (BREEDS-novel), COT cuts about half of the AC error. On natural shift, COT also improves upon AC. On ImageNet natural shift datasets, COT reduces the error from 8.5 to 3.88 compared to AC. COTT, presents the best overall results, surpassing the best current method (ATC) by a notable margin. On synthetic shift benchmarks, COTT is 2-3$\times$ better than ATC-NE, the stronger version of ATC that uses a negative entropy score function. On novel subpopulation shift benchmarks, COTT is at least 4 absolute percent better than ATC-NE. On natural shift benchmarks, however, we observed mixed results. On ImageNet-Natural, COTT is better than ATC-NE while on CIFAR10-Natural, COTT is marginally worse. On WILDS benchmarks, no single method dominantly outperforms others. Note that the WILDS benchmark datasets contain label shift, meaning $P_S(y) \neq P_T(y)$. This leads COTT to overestimate the error on the CivilComments-WILDS. Nonetheless, COTT has the smallest worst-case error of 5.82 compared to ATC-NE's 7.73, demonstrating its robustness even on distribution shifts faced in the wild.

In Figure 4, we use scatterplots to visualize estimations given by different methods, notably AC, ATC, COT, and COTT, where we plot the predicted error against the true error. Ideally, these scattered points should demonstrate a strong linear correlation and closely follow the $y = x$ line. While this is true for COT and COTT, AC and ATC often underestimate, sometimes by a large margin. In Nonliving-26, we can see data points representing shifts whose ATC predicted errors are around 0.1 while true errors are close to 1. These observations corroborate the necessity of leveraging pseudo-label shifts to guard against such catastrophic failures of existing confidence-based methods. We include the scatterplots for all datasets in the supplemental material, where we show that COT and COTT successfully prevent severe underestimation seen in other methods.

## 5 Conclusion

In this work, we proposed COT and COTT, simple yet effective methods leveraging the optimal transport framework to estimate a classifier's performance on out-of-distribution data with only unlabeled samples. Our method is motivated by the observation that existing methods failed to take into account the fact that classifiers often demonstrate pseudo-label shifts when performance drops on an unseen target domain. We formalized this observation by connecting AC and COT under the Wasserstein metric and showing that COT provably provides more conservative estimates by assuming a target marginal distribution. Combining COT with a thresholding strategy that has shown extensive empirical success, we further introduced COTT. On all eleven benchmark tasks, COT and COTT manage to avoid severe underestimation of the model's error seen in current methods. In particular, COTT achieves significantly lower prediction errors than existing methods for most models and datasets (up to 3x better), establishing new state-of-the-art results. We believe our work makes

an important step forward in making performance estimation methods more reliable and theoretically grounded and thus more applicable in real-world applications.

**Limitations:** Despite the strong empirical performance, our method does suffer from several limitations. First, our method is currently only applicable to multi-class classification problems under the assumption that the marginal label distribution must remain constant between the train and test distributions, though as we will show in the supplemental material, our method is still quite robust when the difference between source and target marginal is small. In addition, while we have derived a lower bound for error estimates, we have not currently formulated an upper bound; however, we have not observed overestimating the error to be a common problem empirically. Lastly, the relationship between miscalibration and pseudo-label shift is currently only empirical in nature, not theoretical.

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

# A Deferred Proofs

For readers' convenience, we review the statements of the propositions and corollaries and provide the full proofs below.

## A.1 Proof of Proposition 1

**Proposition 1** ($\hat{\epsilon}_{AC}$-$W_\infty$ Equivalence). *Let $(\mathcal{P}(\mathbb{R}^k), W_\infty)$ be the metric space of all distributions over $\mathbb{R}^k$, where $W_\infty$ is the Wasserstein distance with $c(x, y) = \|x - y\|_\infty$. Then, the estimated error of AC-MC is given by $\hat{\epsilon}_{AC} = W_\infty(\vec{f}_\# P(\vec{c}), P_{pseudo}(\vec{y}))$.*

*Proof.* We first show the following equality. Let $j^* = \arg\max_j \vec{f}_j(x)$

$$1 - \vec{f}_{j^*}(x) = \|\vec{y} - \vec{f}(x)\|_\infty \tag{1}$$

where $\vec{y}_j = \mathbb{1}[j = j^*]$. Let $\vec{f}_{-j^*}(x)$ denote the vector $\vec{f}(x)$ with $j^*$-th element removed. Since for a confidence vector $\|\vec{f}(x)\|_1 = 1$,

$$1 - \vec{f}_{j^*}(x) = \|\vec{f}_{-j^*}(x)\|_1 \geq \|\vec{f}_{-j^*}(x)\|_\infty$$

Therefore, we have obtained the desired Equality 1:

$$\|\vec{y} - \vec{f}_j(x)\|_\infty = \max\{\|\vec{f}_{-j^*}(x)\|_\infty, 1 - \vec{f}_{j^*}(x)\} = 1 - \vec{f}_{j^*}(x)$$

Next, we consider the optimal transport plan between $\vec{f}_\# P(\vec{c})$ and $P_{pseudo}(\vec{y})$. Namely, we show *all* confidence vectors $\vec{f}(x^{(i)})$ are coupled with their one-hot pseudo-labels $\vec{y}^{(i)}$. This can be observed by the fact that the one-hot pseudo-label is the one-hot label that achieves the lowest L-infinity cost, i.e.

$$\|\vec{f}(x^{(i)}) - \vec{y}^{(i)}\|_\infty \leq \|\vec{f}(x^{(i)}) - \vec{y}'\|_\infty, \forall \vec{y}' \in \{0, 1\}^k \cap \Delta^{k-1}$$

Suppose there exist confidence vectors that are not coupled with their one-hot pseudo-labels, then *all* individual costs are suboptimal and the total cost is suboptimal as well, contradicting the assumption that the transport plan is optimal. Therefore,

$$W_\infty(\vec{f}_\# P(\vec{c}), P_{pseudo}(\vec{y})) = \frac{1}{n}\sum_{i=1}^{n}\|\vec{f}(x^{(i)}) - \vec{y}^{(i)}\|_\infty \tag{2}$$

Combining Equality 1 and Equality 2, we obtain the desired relationship between AC error estimate and $W_\infty$ distance

$$\hat{\epsilon}_{AC} = \frac{1}{n}\sum_{i=1}^{n}(1 - \max_j \vec{f}_j(x^{(i)})) = \frac{1}{n}\sum_{i=1}^{n}\|\vec{f}(x^{(i)}) - \vec{y}^{(i)}\|_\infty = W_\infty(\vec{f}_\# P(\vec{c}), P_{pseudo}(\vec{y}))$$

$\square$

## A.2 Proof of Corollary 1

**Corollary 1** ($P_{pseudo}(\vec{y})$ is closest to $\vec{f}_\# P(\vec{c})$). *Let $P'(\vec{y}) \in \mathcal{P}(\{0, 1\}^k \cap \Delta^{k-1})$ be a one-hot label distribution. Then $W_\infty(\vec{f}_\# P(\vec{c}), P'(\vec{y})) \geq W_\infty(\vec{f}_\# P(\vec{c}), P_{pseudo}(\vec{y}))$.*

*Proof.* We first show the following equality, which establishes the relationship between AC accuracy estimate with $W_\infty$ distance:

$$1 - \hat{\epsilon}_{AC} = W_\infty(\vec{f}_\# P(\vec{c}), \delta_0) \tag{3}$$

Since $W_\infty(\vec{f}_\# P(\vec{c}), \delta_0)$ transports $\vec{f}_\# P(\vec{c})$ to $\delta_0$, the optimal transport plan couples every element in $\vec{f}_\# P(\vec{c})$ to 0. For each $x^{(i)}$, its confidence vector $\vec{f}_\# P(\vec{c})$ has a transport cost $\|\vec{f}(x^{(i)}) - 0\|_\infty$. Hence,

$$1 - \hat{\epsilon}_{AC} = \frac{1}{n}\sum_{i=1}^{n}\max_j \vec{f}_j(x^{(i)}) = \frac{1}{n}\sum_{i=1}^{n}\|\vec{f}(x^{(i)}) - 0\|_\infty = W_\infty(\vec{f}_\# P(\vec{c}), \delta_0)$$

With this, our inequality is simply the Triangle Inequality in $(\mathcal{P}(\mathbb{R}^k), W_\infty)$,

$$W_\infty(\vec{f}_\# P(\vec{c}), \delta_0) + W_\infty(\vec{f}_\# P(\vec{c}), P'(\vec{y})) \geq W_\infty(P'(\vec{y}), \delta_0) = 1$$

Combined with Equation 3, we obtain the desired inequality

$$W_\infty(\vec{f}_\# P(\vec{c}), P_{\text{pseudo}}(\vec{y})) = 1 - W_\infty(\vec{f}_\# P(\vec{c}), \delta_0) \leq W_\infty(\vec{f}_\# P(\vec{c}), P'(\vec{y}))$$

$\square$

### A.3  Proof of Proposition 2

**Notations:** Let $\mathcal{C}(\vec{c}) = \{\vec{c}' \in \Delta^{k-1} | \arg\max_j \vec{c}'_j = \arg\max_j \vec{c}_j\}$ be the set of confidence vectors whose one-hot pseudo-labels that match with that of $\vec{c} \in \Delta^{k-1}$. Let $\mathcal{P}(\Delta^{k-1})$ be the set of all distributions of confidence vectors and $\mathcal{P}_c[P_{\text{pseudo}}(\vec{y})] = \{P'(\vec{c}) \in \mathcal{P}(\Delta^{k-1}) | P_{\text{pseudo}}(\vec{y}) = P'(\arg\max_j \vec{c}_j = \arg\max_j \vec{y}_j)\}$ be the set of distributions of confidence vectors that share the same pseudo-label distribution $P_{\text{pseudo}}(\vec{y})$.

$\mathcal{P}_c[P_{\text{pseudo}}(\vec{y})]$ defines an equivalence class for the space of distributions of confidence vectors $(\mathcal{P}(\Delta^{k-1}), W_\infty)$ that share the same pseudo-label distribution $P_{\text{pseudo}}(\vec{y})$. Pictorially, in Figure 1, $\mathcal{P}_c[P_{\text{pseudo}}(\vec{y})]$ represents the line between $\delta_0$ and $P_{\text{pseudo}}(\vec{y})$. On this line, every distribution of confidence vectors shares the same pseudo-label distribution $P_{\text{pseudo}}(\vec{y})$.

To prove Proposition 2, we need the following lemma, which intuitively allows us to change the metric from measuring the distance between two points to the distance between an equivalence class and a point.[*]

**Lemma 1** (Change-of-metric). *Let $\vec{y}, \vec{y}' \in \{0,1\}^k \cap \Delta^{k-1}$ be two one-hot labels. Then the following holds*

$$\inf_{\vec{c} \in \mathcal{C}(\vec{y})} \|\vec{c} - \vec{y}'\|_\infty = 0.5 \times \mathbb{1}[\vec{y} \neq \vec{y}']$$

*Proof.* If $\vec{y} = \vec{y}'$, then we know the optimal $\vec{c} = \vec{y}$

$$\inf_{\vec{c} \in \mathcal{C}(\vec{y})} \|\vec{c} - \vec{y}'\|_\infty = \|\vec{y} - \vec{y}'\|_\infty = 0$$

If $\vec{y} \neq \vec{y}'$, then we proceed by showing equality with two inequalities. First, observe $\{(0.5 + \delta)\vec{y} + (0.5 - \delta)\vec{y}' | \delta \in (0, 0.5]\} \subset \mathcal{C}(\vec{y})$.

$$\inf_{\vec{c} \in \mathcal{C}(\vec{y})} \|\vec{c} - \vec{y}'\|_\infty \leq \inf_{\delta \in (0, 0.5]} \|(0.5 + \delta)\vec{y} + (0.5 - \delta)\vec{y}' - \vec{y}'\|_\infty = \inf_{\delta \in (0, 0.5]} (0.5 + \delta) = 0.5$$

If $\|\vec{c} - \vec{y}'\|_\infty < 0.5$, $\arg\max_j \vec{c}_j = \arg\max_j \vec{y}'_j \neq \arg\max_j \vec{y}_j$, i.e. $\vec{c} \notin \mathcal{C}(\vec{y})$. Therefore, $\inf_{\vec{c} \in \mathcal{C}(\vec{y})} \|\vec{c} - \vec{y}'\|_\infty \geq 0.5$, which further implies $\inf_{\vec{c} \in \mathcal{C}(\vec{y})} \|\vec{c} - \vec{y}'\|_\infty = 0.5$. $\square$

We are now in a position to prove Proposition 2, which follows from the somewhat surprising fact that the left-hand side of the inequality is simply the distance between $\vec{f}_\# P_T(\vec{c})$ and $P_T(\vec{y})$ with a change-of-metric to the metric defined above.

**Proposition 2** (Calibration independent lower bound of COT). *Under the assumption that $P_T(\vec{y}) = P_S(\vec{y})$, we always have $\hat{\epsilon}_{COT} \geq 0.5 W_\infty(P_{pseudo}(\vec{y}), P_T(\vec{y}))$.*

*Proof.* Since $P_T(\vec{y}) = P_S(\vec{y})$,

$$\hat{\epsilon}_{\text{COT}} = W_\infty(\vec{f}_\# P_T(\vec{c}), P_T(\vec{y}))$$

$$= \inf_{\pi(\vec{c}, \vec{y}) \in \Pi(\vec{f}_\# P_T(\vec{c}), P_T(\vec{y}))} \int \|\vec{c} - \vec{y}\|_\infty d\pi(\vec{c}, \vec{y})$$

$$\geq \inf_{\pi(\vec{c}, \vec{y}) \in \Pi(\vec{f}_\# P_T(\vec{c}), P_T(\vec{y}))} \int \inf_{\vec{c}' \in \mathcal{C}(\vec{c})} \|\vec{c}' - \vec{y}\|_\infty d\pi(\vec{c}, \vec{y}) \tag{4}$$

$$= \inf_{\pi(\vec{y}', \vec{y}) \in \Pi(P_{\text{pseudo}}(\vec{y}'), P_T(\vec{y}))} \int \inf_{\vec{c}' \in \mathcal{C}(\vec{y}')} \|\vec{c}' - \vec{y}\|_\infty d\pi(\vec{y}', \vec{y}) \tag{5}$$

---

[*]This is closely related to the Hausdorff distance between sets in a metric space.

Equation 5 follows from the observation that $\mathcal{C}(\vec{c}) = \mathcal{C}(\vec{y}')$ for a confidence vector $\vec{c}$ and its corresponding one-hot pseudo-label $\vec{y}'$. Furthermore, since our new metric $\inf_{\vec{c}' \in \mathcal{C}(\vec{y}')} \|\vec{c}' - \vec{y}\|_\infty$ is only defined up to the equivalence class, replacing each $\vec{c}' \in \mathcal{C}(\vec{c})$ with its pseudo-label $\vec{y}'$ does not change the distance.

Plugging in Lemma 1,

$$
\begin{aligned}
\hat{\epsilon}_{\mathrm{COT}} &\geq \inf_{\pi(\vec{y}',\vec{y}) \in \Pi(P_{\mathrm{pseudo}}(\vec{y}'), P_T(\vec{y}))} \int 0.5 \times \mathbb{1}[\vec{y}' \neq \vec{y}] d\pi(\vec{y}', \vec{y}) \\
&= 0.5 \inf_{\pi(\vec{y}',\vec{y}) \in \Pi(P_{\mathrm{pseudo}}(\vec{y}'), P_T(\vec{y}))} \int \|\vec{y}' - \vec{y}\|_\infty d\pi(\vec{y}', \vec{y}) \\
&= 0.5 W_\infty(P_{\mathrm{pseudo}}(\vec{y}), P_T(\vec{y}))
\end{aligned}
$$

$\square$

## A.4  Tightness of Proposition 2

While Inequality 4 seems loose, our bound is, in fact, tight if no further assumptions on the calibration status of the classifier $\vec{f}$ are made. We need the following lemma that establishes the relationship between pseudo-label shift and the total variation distance between target label distribution and pseudo-label distribution. Note this equivalence only makes sense in the context of measuring $W_\infty$ distance between two one-hot label distributions, but not under other contexts presented in the paper.

**Lemma 2** (Pseudo-label shift is total variation).

$$
W_\infty(P_{pseudo}(\vec{y}), P_T(\vec{y})) = \|P_{pseudo}(\vec{y}) - P_T(\vec{y})\|_{TV}
$$

*Proof.* For two $\vec{y}, \vec{y}' \in \{0,1\}^k \cap \Delta^{k-1}$, the transport cost $c(\vec{y}, \vec{y}') = \mathbb{1}[\vec{y} \neq \vec{y}']$. Then, the standard result on optimal transport [39] gives the desired equality. $\square$

**Corollary 2.**

$$
W_\infty(P_{pseudo}(\vec{y}), P_T(\vec{y})) = 1 - \sum_{\vec{y} \in \{0,1\}^k \cap \Delta^{k-1}} \min\{P_{pseudo}(\vec{y}), P_T(\vec{y})\}
$$

*Proof.*

$$
\begin{aligned}
W_\infty(P_{\mathrm{pseudo}}(\vec{y}), P_T(\vec{y})) &= \|P_{\mathrm{pseudo}}(\vec{y}) - P_T(\vec{y})\|_{TV} \\
&= \frac{1}{2} \sum_{\vec{y} \in \{0,1\}^k \cap \Delta^{k-1}} |P_{\mathrm{pseudo}}(\vec{y}) - P_T(\vec{y})| \\
&= \frac{1}{2} \sum_{\vec{y} \in \{0,1\}^k \cap \Delta^{k-1}} \max\{P_{\mathrm{pseudo}}(\vec{y}), P_T(\vec{y})\} - \min\{P_{\mathrm{pseudo}}(\vec{y}), P_T(\vec{y})\} \\
&= \frac{1}{2} \sum_{\vec{y} \in \{0,1\}^k \cap \Delta^{k-1}} \max\{P_{\mathrm{pseudo}}(\vec{y}), P_T(\vec{y})\} + \min\{P_{\mathrm{pseudo}}(\vec{y}), P_T(\vec{y})\} \\
&\quad - \sum_{\vec{y} \in \{0,1\}^k \cap \Delta^{k-1}} \min\{P_{\mathrm{pseudo}}(\vec{y}), P_T(\vec{y})\} \\
&= \frac{1}{2} \sum_{\vec{y} \in \{0,1\}^k \cap \Delta^{k-1}} P_{\mathrm{pseudo}}(\vec{y}) + P_T(\vec{y}) - \sum_{\vec{y} \in \{0,1\}^k \cap \Delta^{k-1}} \min\{P_{\mathrm{pseudo}}(\vec{y}), P_T(\vec{y})\} \\
&= 1 - \sum_{\vec{y} \in \{0,1\}^k \cap \Delta^{k-1}} \min\{P_{\mathrm{pseudo}}(\vec{y}), P_T(\vec{y})\}
\end{aligned}
$$

$\square$

Finally, we show Proposition 2 is tight by constructing a sequence of distributions of confidence vectors, the limit of which is exactly $0.5 W_\infty(P_{\mathrm{pseudo}}(\vec{y}), P_T(\vec{y}))$ away from $P_T(\vec{y})$.

**Lemma 3** (Proposition 2 is tight)**.**

$$\inf_{P(\vec{c})\in\mathcal{P}_c[P_{pseudo}(\vec{y})]} W_\infty(P(\vec{c}), P_T(\vec{y})) = 0.5 W_\infty(P_{pseudo}(\vec{y}), P_T(\vec{y}))$$

*Proof.* First, we construct the following family of distributions $\{P_\delta(\vec{c})|\delta \in (0, 0.5]\}$, where $P_\delta(\vec{c})$ is the following mixture distribution

$$P_\delta(\vec{c}) = \gamma P_\cap(\vec{y}) + (1 - \gamma)P_\times(\vec{t})$$

where $P_\cap(\vec{y}) = \gamma^{-1}\min\{P_{\text{pseudo}}(\vec{y}), P_T(\vec{y})\}$, $\gamma = \sum_{\vec{y}\in\{0,1\}^k\cap\Delta^{k-1}}\min\{P_{\text{pseudo}}(\vec{y}), P_T(\vec{y})\}$, $P_\times(\vec{t})$ is a distribution supported on $\Delta^{k-1}\cap\{0.5+\delta, 0.5-\delta, 0\}^k$ (i.e. one element in $\vec{t}$ is $0.5+\delta$, another is $0.5-\delta$, and the rest are 0). Additionally, $P_\times(\vec{t}_i = 0.5+\delta) = P_{\text{pseudo}}(\vec{y}_i = 1)$ and $P_\times(\vec{t}_i = 0.5-\delta) = P_T(\vec{y}_i = 1)$. It is easy to check that $P_\delta(\vec{c}) \in \mathcal{P}_c[P_{\text{pseudo}}(\vec{y})]$.

Next, we construct an explicit transport plan $\pi(\vec{c}, \vec{y}) \in \Pi(P_\delta(\vec{c}), P_T(\vec{y}))$. We construct it via the factorization $\pi(\vec{c}, \vec{y}) = P_\delta(\vec{c})\pi(\vec{y}|\vec{c})$, where

$$\pi(\vec{y}|\vec{c}) = \begin{cases} 1 & \text{if } \vec{c} = \vec{y} \text{ or } \langle\vec{c}, \vec{y}\rangle = 0.5 - \delta \\ 0 & \text{otherwise} \end{cases}$$

The cost of this transport plan is therefore

$$\int \|\vec{c} - \vec{y}\|_\infty d\pi(\vec{c}, \vec{y}) = (0.5 + \delta)(1 - \gamma) = (0.5 + \delta)W_\infty(P_{\text{pseudo}}(\vec{y}), P_T(\vec{y}))$$

where the last equality follows from Lemma 2. Taking infimum,

$$\inf_{\delta\in(0,0.5]}\int \|\vec{c} - \vec{y}\|_\infty d\pi(\vec{c}, \vec{y}) = 0.5 W_\infty(P_{\text{pseudo}}(\vec{y}), P_T(\vec{y}))$$

Combining everything so far, we obtain the desired result:

$$0.5 W_\infty(P_{\text{pseudo}}(\vec{y}), P_T(\vec{y})) = \inf_{\delta\in(0,0.5]}\int \|\vec{c} - \vec{y}\|_\infty d\pi(\vec{c}, \vec{y})$$

$$\geq \inf_{\delta\in(0,0.5]} W_\infty(P_\delta(\vec{c}), P_T(\vec{y})) \tag{6}$$

$$\geq \inf_{P(\vec{c})\in\mathcal{P}_c[P_{\text{pseudo}}(\vec{y})]} W_\infty(P(\vec{c}), P_T(\vec{y})) \tag{7}$$

$$\geq 0.5 W_\infty(P_{\text{pseudo}}(\vec{y}), P_T(\vec{y})) \tag{8}$$

Inequality 6 follows from the fact that the optimal transport plan cannot have a greater cost than our explicit plan $\pi$. Inequality 7 is due to the fact that the family of distribution we are considering is a subset of $\mathcal{P}_c[P_{\text{pseudo}}(\vec{y})]$. Inequality 8 is an application of the lower bound 4 which holds for all $P(\vec{c}) \in \mathcal{P}_c[P_{\text{pseudo}}(\vec{y})]$. □

# B  Extended Results

## B.1  Results with Standard Deviation

We show the full experimental results with standard deviation in Table 2.

## B.2  Qualitative Results

We show the qualitative results (scatter plots) in Fig 5

Table 2: Mean Absolute Error (MAE) between the estimated error and ground truth error to compare different methods. The "shift" column denotes the nature of distribution shifts for each dataset. For vision datasets, we reported results for ResNet18 and ResNet50; for language datasets, we reported results for DistilBERT-base-uncased. The results are averaged over 3 random seeds. We highlight the best-performing method. The number in the parentheses denotes the standard deviation.

| Dataset | Shift | Baselines | | | | | | Ours | |
| | | AC | DoC | IM | GDE | ATC-MC | ATC-NE | COT | COTT |
| --- | --- | --- | --- | --- | --- | --- | --- | --- | --- |
| CIFAR10 | Natural | 5.97 (0.10) | 5.38 (0.08) | 5.87 (0.09) | 5.9 (0.15) | 3.38 (0.14) | **3.15** (0.28) | 5.41 (0.09) | 3.33 (0.13) |
| | Synthetic | 9.1 (0.25) | 8.53 (0.28) | 9.26 (0.35) | 8.84 (0.11) | 4.2 (0.38) | 3.37 (0.30) | 2.17 (0.09) | **1.7** (0.26) |
| CIFAR100 | Synthetic | 10.83 (0.08) | 8.76 (0.22) | 12.07 (0.37) | 11.36 (0.25) | 6.8 (0.39) | 6.63 (0.43) | **2.09** (0.27) | 2.59 (0.01) |
| ImageNet | Natural | 8.5 (0.39) | 7.43 (0.41) | 8.62 (0.47) | 5.62 (0.33) | 3.57 (0.46) | 2.6 (0.66) | 3.88 (0.04) | **2.41** (0.11) |
| | Synthetic | 10.34 (0.83) | 9.28 (0.86) | 12.87 (0.77) | 6.54 (0.37) | 1.59 (0.08) | 3.41 (0.53) | 3.24 (0.28) | **1.42** (0.29) |
| Entity13 | Same | 19.63 (2.17) | 19.2 (2.51) | 17.5 (0.90) | 15.37 (1.06) | 8.09 (0.49) | 7.23 (0.49) | 8.47 (0.66) | **2.61** (0.31) |
| | Novel | 29.61 (2.61) | 29.18 (2.95) | 27.22 (1.08) | 24.48 (0.61) | 14.54 (0.95) | 9.49 (0.70) | 15.9 (0.80) | **5.46** (0.75) |
| Entity30 | Same | 16.97 (0.35) | 16.21 (0.36) | 13.56 (2.53) | 13.98 (0.26) | 8.19 (1.07) | 9.08 (0.42) | 5.9 (0.29) | **2.46** (0.65) |
| | Novel | 27.57 (0.06) | 26.81 (0.61) | 23.96 (2.79) | 23.4 (0.1) | 13.46 (2.55) | 8.57 (2.2) | 15.11 (0.38) | **5.94** (1.17) |
| Living17 | Same | 14.84 (3.36) | 14.67 (3.30) | 11.22 (2.06) | 9.94 (0.48) | 4.88 (0.42) | 5.43 (1.06) | 6.25 (1.91) | **2.94** (1.21) |
| | Novel | 29.61 (3.76) | 29.18 (3.71) | 27.22 (3.45) | 24.48 (0.74) | 14.54 (2.87) | 9.49 (3.25) | 15.9 (2.04) | **5.53** (1.93) |
| Nonliving26 | Same | 19.25 (2.45) | 18.43 (3.13) | 16.6 (0.96) | 12.77 (0.85) | 11.18 (2.77) | 9.69 (0.70) | 7.06 (1.17) | **3.34** (0.90) |
| | Novel | 31.37 (2.99) | 30.54 (3.65) | 28.79 (1.47) | 23.37 (0.61) | 19.93 (4.02) | 16.56 (1.28) | 17.8 (1.53) | **10.46** (3.08) |
| Camelyon17-WILDS | Natural | 9.44 (0.50) | 9.44 (0.49) | 10.24 (0.38) | **5.19** (0.44) | 7.73 (0.72) | 7.73 (0.72) | 7.27 (0.57) | 5.71 (0.94) |
| RxRx1-WILDS | Natural | 5.21 (0.26) | 8.44 (0.15) | 8.09 (0.16) | 7.48 (0.26) | 6.53 (0.10) | 6.86 (0.28) | **3.25** (0.16) | 5.82 (0.31) |
| Amazon-WILDS | Natural | 2.62 (0.16) | 2.35 (0.06) | 2.34 (0.06) | 17.04 (0.84) | 1.63 (0.1) | **1.54** (0.11) | 2.43 (0.04) | 2.01 (0.42) |
| CivilCom.-WILDS | Natural | 1.54 (0.23) | 0.96 (0.19) | **0.86** (0.20) | 8.7 (0.14) | 2.3 (0.34) | 2.3 (0.34) | 1.23 (0.05) | 4.68 (0.39) |

## B.3 Correlation Analysis

ProjNorm [41] leverages pseudo labels on the target domain to retrain a copy of the reference model trained on the source domain. The authors show that the difference between the two models' parameters has a strong linear correlation to the true target error. Following the paper's experimental setup, we conducted the correlation analysis on CIFAR10 and CIFAR100 using three architectures, ResNet18, ResNet50, and VGG11. We note that ProjNorm in fact implicitly leverages the assumption that $P_T(y) = P_S(y)$ as this condition holds for both CIFAR10 and CIFAR100. As Fig. 18 of their paper [41] shows, ProjNorm tends to overestimate when label shift exists.

## B.4 Mild Label Shift

We motivate our methods under the assumption of no label shift. In Proposition 2, we showed that the worst-case underestimate of COT is half of the pseudo-label shift. Under mild label shifts, the guarantee for such worst-case underestimation becomes weaker. This can be observed from the following corollary of Proposition 2:

Table 3: Mean Absolute Error (MAE) between the estimated error and ground truth error to compare different methods under mild label shift. The results are averaged over 3 random seeds. We highlight the best-performing method. The number in the parentheses denotes the standard deviation.

| Dataset | Shift | Baselines | | | | | | Ours | |
|---|---|---|---|---|---|---|---|---|---|
| | | AC | DoC | IM | GDE | ATC-MC | ATC-NE | COT | COTT |
| CIFAR10 | Natural | 5.58 (0.26) | 4.99 (0.23) | 5.50 (0.22) | 5.69 (0.04) | 2.76 (0.32) | 2.47 (0.43) | 3.75 (0.28) | **1.68** (0.34) |
| | Synthetic | 8.67 (0.29) | 8.10 (0.31) | 8.82 (0.38) | 8.47 (0.15) | 3.93 (0.38) | 3.13 (0.32) | **2.76** (0.04) | 4.0 (0.30) |
| CIFAR100 | Synthetic | 10.89 (0.15) | 8.85 (0.22) | 12.14 (0.37) | 11.33 (0.23) | 6.93 (0.44) | 6.76 (0.48) | **1.89** (0.30) | 2.81 (0.07) |
| ImageNet | Natural | 8.36 (0.37) | 7.29 (0.42) | 8.46 (0.48) | 5.54 (0.36) | 3.53 (0.48) | 2.47 (0.71) | 3.74 (0.20) | **2.05** (0.26) |
| | Synthetic | 10.26 (0.83) | 9.19 (0.86) | 12.79 (0.77) | 6.50 (0.40) | **1.61** (0.08) | 3.51 (0.52) | 3.03 (0.25) | 1.75 (0.33) |
| Entity13 | Same | 15.50 (0.38) | 14.60 (0.34) | 15.49 (0.22) | 15.18 (1.03) | 8.51 (0.80) | 7.40 (0.57) | 4.59 (0.23) | **3.24** (0.12) |
| | Novel | 24.39 (0.23) | 23.49 (0.19) | 24.56 (0.05) | 23.48 (0.62) | 14.99 (0.82) | 12.45 (0.64) | 11.19 (0.34) | **4.6** (0.38) |
| Entity30 | Same | 15.46 (0.70) | 13.93 (0.65) | 15.55 (0.74) | 13.83 (0.27) | 8.80 (0.64) | 8.26 (0.83) | 4.75 (0.29) | **2.16** (0.15) |
| | Novel | 25.98 (0.53) | 24.45 (0.46) | 26.72 (0.68) | 23.28 (0.14) | 15.56 (0.56) | 13.21 (0.90) | 13.96 (0.17) | **7.07** (0.27) |
| Living17 | Same | 11.38 (0.67) | 10.90 (0.48) | 11.83 (1.31) | 9.85 (0.41) | 4.46 (0.31) | 4.39 (0.18) | 4.40 (0.34) | **2.71** (0.81) |
| | Novel | 25.72 (0.46) | 25.13 (0.76) | 26.32 (1.98) | 21.61 (0.68) | 14.09 (2.31) | 11.48 (1.99) | 16.94 (0.78) | **9.31** (1.79) |
| Nonliving26 | Same | 16.28 (0.37) | 14.48 (0.29) | 15.69 (0.19) | 12.88 (0.84) | 9.63 (0.43) | 9.69 (0.66 | 5.33 (0.73) | **2.18** (0.23) |
| | Novel | 27.93 (0.08) | 26.13 (0.25) | 27.76 (0.29) | 23.25 (0.69) | 18.15 (0.38) | 16.08 (0.38) | 15.66 (0.45) | **8.71** (0.42) |

Table 4: Coefficients of determination ($R^2$) and rank correlations ($\rho$) to measure the linear correlation between a method's output quantity and the true target error (the higher the better). COT achieves superior performance than all existing methods across different models and datasets.

| Dataset | Network | AC | | Entropy | | GDE | | ATC | | ProjNorm | | COT | |
|---|---|---|---|---|---|---|---|---|---|---|---|---|---|
| | | $R^2$ | $\rho$ | $R^2$ | $\rho$ | $R^2$ | $\rho$ | $R^2$ | $\rho$ | $R^2$ | $\rho$ | $R^2$ | $\rho$ |
| CIFAR10 | ResNet18 | 0.825 | 0.980 | 0.862 | 0.982 | 0.842 | 0.981 | 0.875 | 0.987 | 0.947 | 0.988 | **0.996** | **0.998** |
| | ResNet50 | 0.950 | 0.995 | 0.949 | 0.995 | 0.959 | 0.995 | 0.885 | 0.989 | 0.936 | 0.989 | **0.993** | **0.996** |
| | VGG11 | 0.710 | 0.938 | 0.762 | 0.958 | 0.723 | 0.948 | 0.548 | 0.851 | 0.756 | 0.949 | **0.994** | **0.993** |
| | Average | 0.828 | 0.971 | 0.858 | 0.978 | 0.841 | 0.975 | 0.769 | 0.942 | 0.880 | 0.975 | **0.994** | **0.996** |
| CIFAR100 | ResNet18 | 0.943 | 0.987 | 0.932 | 0.984 | 0.950 | 0.988 | 0.927 | 0.985 | 0.969 | 0.974 | **0.995** | **0.997** |
| | ResNet50 | 0.957 | 0.987 | 0.948 | 0.984 | 0.962 | 0.989 | 0.955 | 0.991 | 0.982 | 0.991 | **0.992** | **0.996** |
| | VGG11 | 0.794 | 0.959 | 0.821 | 0.973 | 0.870 | 0.978 | 0.736 | 0.975 | 0.653 | 0.849 | **0.996** | **0.997** |
| | Average | 0.898 | 0.978 | 0.900 | 0.980 | 0.927 | 0.985 | 0.873 | 0.984 | 0.868 | 0.938 | **0.994** | **0.997** |

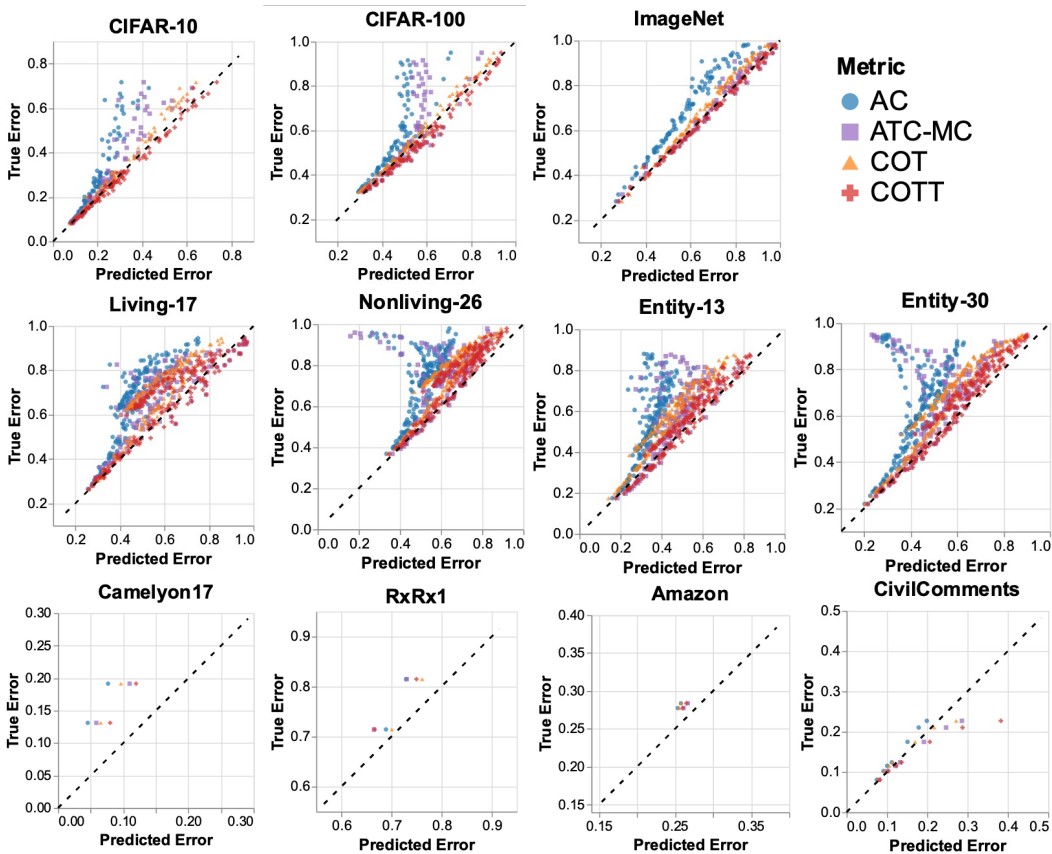

Figure 5: Qualitative results for AC, ATC, COT, and COTT. In these scatterplots, the x-axis is the target error estimate and the y-axis is the ground truth target error. Accurate estimates should be close to $y = x$ (dashed black line). We can see that for all datasets, COT and COTT avoid the severe underestimation seen on ATC.

**Corollary 3** (Calibration independent lower bound of COT under *mild* label shift)**.**

$$\hat{\epsilon}_{COT} \geq 0.5 W_\infty(P_{pseudo}(\vec{y}), P_T(\vec{y})) - W_\infty(P_S(\vec{y}), P_T(\vec{y}))$$

*Proof.* By Triangle Inequality in $(\mathcal{P}(\mathbb{R}^k), W_\infty)$,

$$W_\infty(\vec{f}_\# P_T(\vec{c}), P_S(\vec{y})) + W_\infty(P_S(\vec{y}), P_T(\vec{y})) \geq W_\infty(\vec{f}_\# P_T(\vec{c}), P_T(\vec{y}))$$

Combined with Proposition 2, we obtain the desired result. $\qquad\square$

As the label shift increases, we have a weaker guarantee of the worst-case underestimation error of COT as long as $W_\infty(P_S(\vec{y}), P_T(\vec{y})) \leq 0.5 W_\infty(P_{\text{pseudo}}(\vec{y}), P_T(\vec{y}))$. However, we perform additional controlled experiments which suggest our methods remain to be the most performant despite the theoretical guarantee is not as strong as the case without label shift.

To simulate mild label shift for datasets with $P_S(\vec{y}) = P_T(\vec{y})$, we first calculate the original target marginal and then sample the shifted target marginal from a Dirichlet distribution as in [14] with a parameter $\alpha = 50$. The parameter $\alpha$ controls the severity of the label shift, and a smaller $\alpha$ means a larger label shift. Concretely, let the shifted target marginal be $P_{\tilde{T}}(\vec{y})$. Then $P_{\tilde{T}}(\vec{y}) \sim \text{Dir}(\beta)$ where $\beta_{(\vec{y})} = \alpha \cdot P_T(\vec{y})$. Finally, based on $P_{\tilde{T}}(\vec{y})$, we sample a new set of test samples for which we estimate the performance. We conducted this mild label shift experiment for CIFAR10, CIFAR100, ImageNet, Living17, Nonliving26, Entity13, and Entity30 as these datasets have the same source and target marginal. We showed the results in Table 3. As we can see, our methods still dominate existing methods under this relaxed condition.

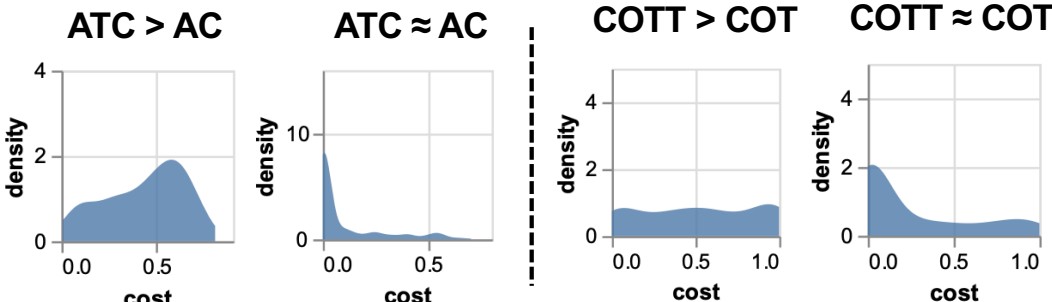

Figure 6: We demonstrate cases where using thresholding improves over taking averages. The x-axis denotes the max norm between a confidence vector and the corresponding one-hot label. For AC and ATC-MC, the corresponding label is always the argmax of the confidence vector as mentioned in section 2.3. For COT and COTT, the corresponding label is the one matched via optimal transport. We observe that thresholding improves over averaging when the cost distribution is less concentrated around 0, which corresponds to situations where the model is very confident on most samples.

### B.5 When does thresholding improve over averaging?

In this section, we provide some intuitions on when using a threshold provides better estimates than taking the average. From Fig. 6, we show that thresholding yields larger and more accurate error estimates when the cost distribution on the OOD data is more spread out and less concentrated around 0. By contrast, when the cost distribution is mostly near 0, thresholding leads to similar estimates as averaging. Interestingly, even on OOD data where the model has very low performance, there is still a decent amount of samples whose cost is near 0. Thus, when taking the average, we will end up with a smaller value which suggests a low error. In these cases, thresholding will give larger error estimates than averaging.

## C  Datasets

**CIFAR10:**   The synthetic shifts included 19 common visual corruptions across 5 levels of severity from [19]. The natural shift is CIFAR10-V2 [33].

**CIFAR100:**   The synthetic shifts included 19 common visual corruptions across 5 levels of severity from [19].

**ImageNet:**   The synthetic shifts included 19 common visual corruptions across 5 levels of severity from [19]. The natural shifts include 4 datasets from ImageNet-V2 [34] and ImageNet-Sketch [40].

**BREEDS:**   The BREEDS benchmark contains 4 datasets, Living-17, Nonliving26, Entity13, Entity30. For each of the datasets, the same subpopulation shifts include the corrupted versions of the test set with the same subpopulation; the novel subpopulation shifts include the clean as well as corrupted versions [19] of the test set with novel subpopulation.

**WILDS:**   For all WILDS datasets, we used the official OOD datasets provided in their paper [25].

## D  Experiment Setup

We performed training in PyTorch [31], and we used RTX 6000 Ada GPUs.

For datasets without an official validation set, we randomly sampled a subset of the official training set as the validation set to perform calibration and learn thresholds for ATC and COTT. We trained 3 models for each dataset with random seeds $\{0, 1, 10\}$.

**CIFAR10 and CIFAR100:**   We reserved 10000 images from the training set as the validation set. We trained ResNet18 from scratch, using SGD with momentum equal to 0.9 for 300 epochs. We set

weight decay to $5 \times 10^{-4}$ and batch size to 200. We set the initial learning rate to 0.1 and multiply it by 0.1 every 100 epochs.

**ImageNet:** We reserved 50000 images from the training set as the validation set. We used ResNet50. While ImageNet pretrained weights are available in PyTorch, we needed multiple ones trained using different initializations. Due to limited computation resources, we reused the upper layer weights but reinitialized the last layer with different random seeds. We finetuned the whole model using Adam [23] with a batch size of 64 and a learning rate of $10^{-4}$, for 10 epochs.

**BREEDS:** We used the intersection set of images that are both in the ImageNet validation images we set aside and the BREEDS dataset as the validation set. For all BREEDS datasets (Living17, Nonliving26, Entity13, Entity30), we trained ResNet50 from scratch.

For Living17 and Nonliving26, we used SGD with weight decay of $10^{-4}$ and batch size of 128. We trained for 450 epochs. We set the initial learning rate to 0.1 and multiplied it by 0.1 every 150 epochs.

For Entity13 and Entity30, we used SGD with weight decay of $10^{-4}$ and batch size of 128. We trained for 300 epochs. We set the initial learning rate to 0.1 and multiplied it by 0.1 every 100 epochs.

**Camelyon17-WILDS:** We used the `id_val` group as the validation set. We fine-tuned ImageNet pretrained ResNet50 using SGD with momentum of 0.9, weight decay of $5 \times 10^{-4}$, and batch size of 32, for 5 epochs.

**RxRx1-WILDS:** We used the `id_text` group as the validation set. We followed [25] to fine-tune an ImageNet pretrained ResNet50. We used Adam with weight decay of $10^{-5}$ and batch size of 75, for 90 epochs. We increased the learning rate from 0 to $10^{-4}$ linearly for the first 10 epochs and decayed it following a cosine learning rate schedule.

**Amazon-WILDS:** We used the `id_val` group as the validation set. We followed [25] to fine-tune a DistilBERT-base-uncased model [35]. We used AdamW [28] with weight decay of $10^{-2}$, learning rate of $10^{-5}$, and batch size of 8, for 3 epochs. We set the maximum number of tokens to 512.

**CivilComments-WILDS:** We used the `val` group as the validation set. We followed [25] to fine-tune a DistilBERT-base-uncased model [35]. We used AdamW [28] with weight decay of $10^{-2}$, learning rate of $10^{-5}$, and batch size of 16, for 5 epochs. We set the maximum number of tokens to 300.

