# OpenReview forum: "Characterizing Out-of-Distribution Error via Optimal Transport"
_NeurIPS.cc/2023/Conference — NeurIPS 2023 poster_

### Official Review · Reviewer_chpC · 2023-07-06

**Soundness:** 2 fair
**Presentation:** 3 good
**Contribution:** 2 fair
**Rating:** 5
**Confidence:** 3

**Summary:**

In this work, the authors aim to predict model’s performance on out-of-distribution (OOD) data without access to labels during testing. Specifically, they identify pseudo-label shift, which gauges  the difference between the predicted and the true OOD label distributions, to indicate the under-estimation issue suffered by the existing methods. Based on the obervision, they propose Confidence Optimal Transport (COT) and an empirically-motivated variant of COT, Confidence Optimal Transport with Thresholding (COTT), for more robust OOD error estimation. Experiments are conducted on 10 datasets with various types of distribution shift to verify the effectiveness of the proposed method.

**Strengths:**

1. The motivation in this work is clear. The authors explain the under-estimation issue encountered by previous studies through the lens of pseudo-label drift, and propose a method leveraging optimal transport theory based on the observision.

2. The code is available with the submission which improves the reproducibility index of the paper.

**Weaknesses:**

1. Since this paper contains many mathematical symbols, the meaning of some symbols is confused, which increases the difficulty of understanding. For example, in Section 2.1, you mention to predict the error $\epsilon = 1-\alpha$, but later the predicted error is denoted as $\hat{\epsilon}$. Comparing Figure 1 with Section 3.1, I find the assumption changes from $P_T(\vec{y}) \approx P_S(\vec{y})$ to $P_T(\vec{c}) \approx P_S(\vec{y})$. Is the meaning of $P_T(\vec{y})$ equivalent to that of $P_T(\vec{c})$? In addition, in Section 3.1, what is the meaning of $\hat{P}_S(\vec{y})$ and $\hat{P}_T(\vec{y})$? I suggest that the authors explain them after every equation and use harmonized symbols.

2. The relationships between pseudo-label drift and the true target error is not clear. In Section 2.4, you briefly mention that the pseudo-label shift is a lower bound of the true target error of the model. Can you explain the property in details? Specifically, I prefer to know the relationships between your method (or miscalibration) and the true target error. Can you give some insights?

3. This work is potentially vulnerable to two cases: 1) imbalanced test set.  2) The label space of the test set is only the subset of that of the training set.

**Questions:**

See the weaknesses.

**Limitations:**

The author provides the limitations of their work. It would be better if the authors discuss more cases such as the two cases mentioned in the main review.

---

> ### Author Rebuttal · Authors · 2023-08-10
>
> We thank the reviewer for their valuable review of our work! In particular, we are excited that our motivation reads clear to the reviewer and our code submission alongside this contribution improves the reproducibility of our method.
>
> ### On Weakness 1
>
> We thank the reviewer for pointing out the confusion and will clarify them below. We have updated our paper to incorporate the suggestions and corrections suggested by the reviewer to make it more readable.
> - Regarding the notation for test error, we would like to clarify that ${\epsilon}$ is the true test error (that is assumed to be inaccessible and must be estimated/predicted), while $\hat{\epsilon}$ is the predicted test error (given by an error estimation algorithm), as defined in line 84 of our paper. Our goal is to output $\hat{\epsilon}$ as close to ${\epsilon}$ as possible.
> - The $\vec{c}$ in line 188 should be $\vec{y}$.
> - In Section 3.1, the hat notations denote the empirical distributions, that is the distributions over the finite samples.
>
> ### On Weakness 2
>
> We thank the reviewer for asking for more details and intuitions on pseudo-label shift. We provide a more detailed explanation below and have updated our paper correspondingly.
> Pseudo label shift is the lower bound of the true error because as we mentioned in Section 2.4, it is the Wasserstein distance between $P_{pseudo} (y)$ and $P _T (y)$, defined as the minimum transport cost over all transport maps between these two distributions. Meanwhile, the true error corresponds to a specific transport map: where the pseudo label of each sample is mapped to its true label, so that is why it is lower bounded by the pseudo-label shift. For the relationship between our method and true target error, we would like to direct the reviewer to Proposition 2 (line 194) of our paper, where we established that the error estimate of COT is never smaller than half the pseudo-label shift, which is the lower bound of true target error.
>
> ### On Weakness 3
>
> We thank the reviewer for pointing out potential vulnerable cases and will address them below. We have incorporated the discussion on these two cases in our paper.
> - Imbalanced test set: Theoretically, our method is agnostic to different label distributions, assuming no/mild label shift between in-distribution and out-of-distribution label distributions. Empirically, We would like to note that RxRx1-WILDS, Amazon-WILDS, and CivilCom.-WILDS are all imbalanced datasets and COT/COTT provides competitive performance on these datasets.
> - The label space of the test set is only the subset of that of the training set: If a big fraction of the training set labels do not show up in the test set, this will cause a large label shift, which would lead our methods to overestimate the error. However, this setting is outside the scope of this paper as we currently consider no/mild label shift between in-distribution and out-of-distribution test sets. In the future, we plan to explore the use of Partial Optimal Transport for this more challenging setting.

---

> > ### Comment · Reviewer_chpC · 2023-08-13
> >
> > Dear Authors,
> >
> > Thank you for your detailed response to my reviews. My concerns have been addressed, so I chose to raise my score.
> >
> > Best regards,
> >
> > Reviewer chpC

---

### Official Review · Reviewer_dn4Z · 2023-07-06

**Soundness:** 3 good
**Presentation:** 2 fair
**Contribution:** 3 good
**Rating:** 6
**Confidence:** 3

**Summary:**

Authors develop a methodology for estimating the accuracy of a classifier in the case where the distribution of the test set is different from the one used in the training data. Their methodology is based on computing a certain Wasserstein distance between suitable distributions on labels. Authors argue this definition is robust to pseudo label shift and validate their claims through extensive experimental validation.

**Strengths:**

StrengthsI think overall it is a good, strong paper. It addresses an important problem and it provides a highly innovative perspective on that problem. Experiments are compelling and make a strong point that the method is able to combat otherwise pervasive underestimation of error for of out of distribution samples.

**Weaknesses:**

The main weakness is the lack of sufficient clarity. The entire argument is subtle and required a super clear explanation. Unfortunately, the way the manuscript is written does not help much. Another weakness is that while this is promoted as an OT based method, the theoretical results (while seemingly correct) don’t enlighten much about what the role of Optimal transport is.




**Questions:**

1)Proposition 2 is nice but it is unclear how such lower bound can be ever applied. In other words, the estimated error could be 1 always and the result is useless. Authors should elaborate more on the significance of this result.

2)Similarly, Corollary 1 is helpful (line 189) but again, doesn't say the proposed method is always good. What AC got it just right while COT overestimates the error? The fact that there is a GAP between great experimental results and these somewhat weak theoretical results is a bit too bad and authors (although perhaps not here) should aim to clarify this.

3)Authors may try to better explain the role of the Wasserstein infinity distance here. Is there anything special about this metric space that makes things work or it is just a mathematical convenience? The proofs are based on geometrical intuitions, but is there anything special about the Wasserstein geometry here?

4) It would be great if the authors could elaborate more on why the Pseudo label shift is the right quantity to focus on.  By line 147 I am worried that the observed correlation showed in the experiments could be an artifact of the fact that the pseudo label shift is bounded by \epsilon. How can this be ruled out?

5) The definition of Pppseuudi in line 95 page 3 is awkward as it is not immediately clear whether it refers to domain X or Y.

6) inline equation in line 188 page 5 seems wrong. Could you please correct?

**Limitations:**

n.a.

---

> ### Author Rebuttal · Authors · 2023-08-10
>
> We thank the reviewer for their thorough review and are encouraged to hear that our work was perceived as a good and strong contribution, providing a highly innovative perspective on the problem of OOD detection. Further, we appreciate the reviewer’s comment that our experiments are compelling in supporting our claims. In the following paragraphs, we will provide further clarifications on our method, which we have also incorporated into the updated version of our manuscript.
>
> ### On Weakness:
>
> We thank the reviewer for the advice to improve the writing of our paper to make our argument stronger. We have added more explanations and details to make our messages clearer. On role of OT, we think Figure 1 can provide some intuition. Essentially, using OT, our method can provide an error estimate that is aware of the assumed label distribution. From here, we proved formal guarantees of our estimation, i.e., COT provides provably more conservative estimates than AC, and COT is never lower than half the pseudo-label shift, which is the lower bound of true target error. We hope this gives a clearer picture of our work but welcome any follow-up questions.
>
> ### On Question 1 and 2:
> The reviewer insightfully pointed out the fact that both Corollary 1 and Proposition 2 are lower bounds of COT and there is no upper bound guarantee. These lower bounds are useful because NNs tend to be overconfident [1], resulting in overly optimistic performance estimates for existing confidence-based methods. Given this observation, lower bounds are actually more useful than upper bounds.  On the other hand, we believe upper bounds are difficult to obtain and require a much deeper theoretical understanding of issues like calibration. While more calibration methods have been proposed over the years, the reason why and when neural networks are miscalibrated is still not well understood [2].
>
> We admit that it is indeed possible for AC to be just right and COT to overestimate. However, we rarely observe this phenomenon empirically, because when the model makes a lot of mistakes, very low confidence is required to have a large overestimation of the correct set, but we know neural nets are more prone to be overconfident than not [1]. As a result, AC almost always underestimates the error. Thus, we think the two lower bounds of COT we proved are helpful in explaining the success of our methods on a large number of datasets. We hope this explanation helps clarify things but welcome any follow-up questions during the discussion period.
>
> ### On Question 3:
>
> Role of Wasserstein infinity distance: the reviewer’s observation is astute. The infinity norm is the most natural distance for AC-MC, which computes the maximum confidence in a confidence vector. This is precisely the definition of the infinity norm of a confidence vector. The infinity norm enables us to shine a new light on AC that is previously unexplored. Bringing AC into the abstract geometric framework allows us to obtain results previously unattainable:
> 1. Understanding the well-known failure mode in AC-MC via projection optimality, an inherent geometric notion.
> 2. Design COT and provide a guarantee that safeguards the method from similar issues. While similar results may be obtainable with enough effort through other perspectives, we believe our novel geometric insights are interesting to the community.
>
> As a side note, it is a simple exercise to show that the L-1 norm and the L-infinity norm are equivalent to COT. However, the L-1 norm does not provide a unified perspective for AC-MC, which is why the L-infinity norm is important here.
>
> ### On Question 4:
>
> The reasons why we consider the pseudo-label shift are the following:
> - When source and target label marginals are the same, pseudo-label shift is the lower bound of the true error, because as we mentioned in Section 2.4, it is the Wasserstein distance between $P_{pseudo} (y)$ and $P_T (y)$, defined as the minimum transport cost over all transport maps between these two distributions. Meanwhile, the true error corresponds to a specific transport map: where the pseudo label of each sample is mapped to its true label, so that is why it is lower bounded by the pseudo-label shift.
> - Under the assumption that the target label distribution is close to the source one, pseudo-label shift is a quantity that we can actually estimate in practice, and this quantity provides us with additional information about the true target error.
> - It also has an intuitive triangular relationship with AC-MC and COT, as illustrated in Figure 1. Using the additional information about the target label distribution allows us to get a more accurate prediction when the model is miscalibrated, as we discussed in Sections 2.3 and 2.4.
>
> How to Interpret the correlation between pseudo-label shift and AC-MC’s estimation error:
> - The reviewer is correct that the pseudo-label shift lower bounds the true error. Assuming reference to Figure 2, we would like to clarify that the y-axis is AC-MC’s estimation error instead of the true error. We hope this helps clarify things but if not, we welcome any follow-up questions during the discussion period.
>
> ### On Question 5 and 6:
>
> We thank the reviewer for their writing suggestions. The $\vec{c}$ in line 188 should be $\vec{y}$. We have incorporated them and corrected the typos in our paper.
>
> ### References:
>
> [1] Ovadia, Yaniv, et al. "Can you trust your model's uncertainty? evaluating predictive uncertainty under dataset shift." Advances in neural information processing systems 32 (2019).
>
> [2] Wang, Cheng. "Calibration in Deep Learning: A Survey of the State-of-the-Art." arXiv preprint arXiv:2308.01222 (2023).

---

> > ### Comment · Reviewer_dn4Z · 2023-08-15
> > **Thank you**
> >
> > Thanks a lot for your in-depth rebuttal. All my concerns have been addressed and I have maintained my already positive assessment of this paper.

---

### Official Review · Reviewer_1ruP · 2023-07-07

**Soundness:** 3 good
**Presentation:** 2 fair
**Contribution:** 3 good
**Rating:** 5
**Confidence:** 2

**Summary:**

This paper proposed Confidence Optimal Transport (COT) and a variant, Confidence Optimal Transport with Thresholding (COTT) based on pseudo-label shift to characterize the error between predicted and true OOD label distribution.

**Strengths:**

1. The idea of using pseudo-label shift to characterize error is natural and straight-forward.
2. Experiment results are reasonable.

**Weaknesses:**

1. This work does not have sufficient theoretical support for the method.
2. I am not fully convinced that the main assumption of same source label distribution and target label distribution is reasonable.

**Questions:**

1. I don't understand the assumption: "the target label distribution is close to the source label distribution" and think this assumption conflicts with the purpose of detecting OOD error --the test set should be assumed to behave differently from the training set and that's where the error come from. I wonder if authors could help clarify this assumption.
2. Figure 2 and 3 do not seem clear to me, could authors elaborate more on the purpose of including types of corruption in the figure?

**Limitations:**

Yes, the authors have addressed their limitations.

---

> ### Author Rebuttal · Authors · 2023-08-10
>
> We thank the reviewer for the thoughtful feedback and for the recognition that our idea is natural and straightforward, and our experiment results are reasonable. We address concerns and questions below.
>
> ### On Weakness 1
>
> Previous works have established that no theoretical guarantee is possible for OOD error estimation on arbitrary distribution shifts (See section 3.1 in Garg et al. 2022 [1] ). Therefore, assumptions are essential to make progress on theoretical guarantees. We believe that one significance of this work is that we specify a specific assumption (no/mild label shift) that, when held true, makes our method more desirable than other methods. Most previous methods rely on the assumption that the model is well-calibrated. However, modern neural networks are well known to be overconfident and miscalibrated. While calibration methods such as temperature scaling help in the in-distribution setting, they fail miserably under distribution shift. See Ovidia et al. 2019 [2] for a comprehensive study. Under the no/mild label shift assumption, we additionally showed that COT provides provably more conservative estimates than AC, and established the relationship between COT and pseudo-label shift. We would appreciate it if the reviewer could elaborate on what their ideal theoretical support looks like so we could further discuss if such desired results are attainable.
>
> ### On Weakness 2 and Question 1
>
> To start with, we would like to clarify that we mainly focus on covariate shifts in this work. For example, different hospitals might use different stain colors for their pathology samples. This will have the effect that their images look very different, but their respective label marginals (distribution of classes of diseases) should be very similar. For another example, when a self-driving car operates on rainy days, the images it collects might be very different from ones from sunny days, but the occurrence of objects, for example, traffic lights, stop signs, etc should still be the same. We hope these examples help to clarify why the assumption that the source and label distributions are the same does not defeat the purpose of estimating OOD errors. Additionally, building upon our response regarding weakness 1, since some assumptions are required, we have chosen an assumption that is frequently satisfied, theoretically convenient to work with, and produces strong empirical results. We again thank the reviewer for raising this concern over our assumption. We have updated our paper to explicitly state the distribution shifts we aim to tackle.
>
> ### On Question 2:
>
> We thank the reviewer for pointing out their confusion over Figure 2 and 3 and have updated the captions for them to make it clear.  For Figures 2 and 3, we showed the corruption types and severities to provide readers with the specific distribution shifts we test on in CIFAR-10-C and CIFAR-100-C datasets.
>
> ### References
>
> [1] Garg, S. and Balakrishnan, S., 2022. Leveraging Unlabeled Data to Predict Out-of-Distribution Performance. ICLR.
>
> [2] Ovadia, Yaniv, et al. "Can you trust your model's uncertainty? evaluating predictive uncertainty under dataset shift." Advances in neural information processing systems 32 (2019).

---

> > ### Comment · Reviewer_1ruP · 2023-08-16
> >
> > I thank the authors for addressing my concerns and decide to stick to my rating as 5.

---

### Official Review · Reviewer_1Jqy · 2023-07-14

**Soundness:** 3 good
**Presentation:** 3 good
**Contribution:** 3 good
**Rating:** 7
**Confidence:** 3

**Summary:**

The paper proposes to compute OOD error estimates using a Wassertein distance between the target label distribution and the distribution of confidence vectors. This is in contrast to AC-MC which compare the pseudo-label distribution and the confidence distribution. The paper motivates COT (confidence optimal transport) and COT - threshold to handle outliers and shows they are useful metrics in understanding OOD error. The paper does a variety of experiments to evaluate how the proposed methods do in fact estimate OOD error bettter than existing work.

**Strengths:**

1. The experiments in the paper show clear advantage of COT and COTT for predicting OOD error using unlabelled samples.
2. The presentation in the paper is clear, barring some organization issues.
3. The paper tackles the important problem of estimating OOD error without labels and the proposed method is not very complicated.

**Weaknesses:**

1. The paper does not specify the kind of distribution shift under which the method works; causal shift, anticausal shift, covariate etc.. For example, there are distribution shifts where p_S(X)  = p_T(X), which, given good prediction, would mean small COT and COTT (because confidences are concentrated around the correct label). Could the authors explain how this may not be an issue? Better, could the authors evaluate the methods on a dataset like ColoredMNIST with a flipped spurious correlation between train and test.


2. The writing could be improved by introducing COT and COTT before showing plots with them.

3. I'm a little fuzzy on the intuition of which AC-MC does not work as well as COT. Is it that the confidence distribution becomes more diffuse with distribution shift and AC-MC does not scale with that "diffusion" because the pseudo-label distribution also depends on the confidence distribution, but COT being that it looks at the $p_T(Y)$, breaks that dependennce?

**Questions:**

See weaknesses.

**Limitations:**

In one sense. The weaknesses section outlines the issue of understanding what kind of OOD the method is really testing.

---

> ### Author Rebuttal · Authors · 2023-08-10
>
> We thank the reviewer for their thoughtful feedback and for the compliment that the experiments demonstrate clear advantages of our method, the presentation of the paper is clear, and we tackle an important problem of estimating OOD error without a complicated solution. We would like to address your questions and concerns as follows:
>
> ### On Weakness 1
>
> We thank the reviewer for this great point, and have added clarifications about the shift we studied in the updated version our paper. To clarify, we mainly experimented with covariate shift datasets. For cases where $P _{S} (X)$ = $P _T (X)$, this means we are essentially estimating the model’s performance on in-distribution data. Since we calibrate the model using temperature scaling on the in-distribution validation set, as mentioned in Section 4.2 of our paper, the confidence scores will be close to the in-distribution accuracy on average. As a result, the particular case brought up by the reviewer would not be an issue.
>
> Per the reviewer’s suggestion, we performed additional experiments on the ColorMNIST dataset. We found that COT/COTT provides consistently strong performance as well. Please refer to the PDF file attached to our global response for more detailed experimental results.
>
> ### On Weakness 2
>
> We thank the reviewer for this great suggestion! We have updated the paper accordingly.
>
>
> ### On Weakness 3
>
> The reviewer’s intuition on why AC-MC does not work as well as COT sounds mostly correct to us. Here is a more detailed elaboration on this intuition. Modern neural networks are well known to be overconfident and miscalibrated. While calibration methods such as temperature scaling help in the in-distribution setting, they fail miserably under distribution shift. See Ovidia et al. 2019 [1] for a comprehensive study. Now, AC-MC by definition only works when the calibration is good, which as we mentioned is a poor assumption in practice under OOD settings. This corresponds to the reviewer’s intuition where the confidence becomes “diffused” but not enough, due to overconfidence and failure of OOD calibration. By contrast, COT is able to adjust for these overconfident mispredictions using the assumed label distribution
>
> To provide a more concrete example, a model might have 80% accuracy on in-distribution data and since it’s calibrated its average confidence (AC) is around 80%. Now, on the OOD data, it can have an accuracy of 50% but AC of 70% due to overconfidence. But if pseudo-label shift exists on this OOD data, which we empirically observed to be true in extensive experiments, COT will be able to leverage this information to provide a more accurate estimate of the performance.
>
> ### References
>
> [1] Ovadia, Yaniv, et al. "Can you trust your model's uncertainty? evaluating predictive uncertainty under dataset shift." Advances in neural information processing systems 32 (2019).

---

> > ### Comment · Reviewer_1Jqy · 2023-08-18
> >
> > Thanks for the response. I have updated my score. I had one major concern left that another reviewer points out: the assumption that " the target label distribution is close to the source label distribution". I am inclined to not consider this a very serious issue after 1) the authors pointed out their main concern is covariate shift and also that label shift correct is well-studied and could be applied orthogonal to the method. Could the authors comment on my thinking here and write their own response to the concern that " the target label distribution is close to the source label distribution" is a strong assumption?

---

> > > ### Author Response · Authors · 2023-08-19
> > > **Reply to Reviewer 1Jqy's Inquiry**
> > >
> > > We thank you for acknowledging our rebuttal and for raising the score! We are glad that you requested further clarifications of our assumption that “the target label distribution is close to the source label distribution”, as this assumption is an important part of our paper.
> > >
> > > Your thoughts on the assumption make sense to us. We also think that for covariate shifts, assuming the target label distribution is close to the source label distribution is a natural one. However, while we do agree that label shift correction is well-studied, we would like to note that these correction methods [1] assume there are no covariate shifts, i.e., p(x|y) does not change. Thus, we do not think we can directly use existing methods to relax our assumption. While this is unfortunate, we think this also helps to support our claim that finding the right assumption is critical in making progress on tackling distribution shifts.
> > >
> > > We would address the concern by pointing out: 1) the assumption that “the target label distribution is close to the source label distribution” is a natural assumption for covariate shifts as we observe it to be true in various real-world benchmark datasets we experimented with. 2) the assumption is a powerful one that, when held true, allows us to compute pseudo-label shift, a lower bound of true error, and to propose COT/COTT that provides strong empirical performance and some theoretical guarantee. 3) proposing this assumption has great value, as the community currently working on the problem of estimating OOD error finds it challenging to specify under what circumstances their proposed methods are guaranteed to work. Confidence-based methods mainly rely on the model being well-calibrated, which is rarely true on OOD datasets [2] and can be extremely unreliable as shown in our paper. 4) we plan to work on methods to relax this assumption in the future to make COT/COTT robust to even extreme distribution shifts.
> > >
> > > Thanks again for your reply. Please let us know if our reply addressed your concerns.
> > >
> > > [1] Lipton, Zachary, Yu-Xiang Wang, and Alexander Smola. "Detecting and correcting for label shift with black box predictors." International conference on machine learning. PMLR, 2018.
> > >
> > > [2] Ovadia, Yaniv, et al. "Can you trust your model's uncertainty? evaluating predictive uncertainty under dataset shift." Advances in neural information processing systems 32 (2019).

---

### Official Review · Reviewer_HB65 · 2023-07-22

**Soundness:** 3 good
**Presentation:** 3 good
**Contribution:** 3 good
**Rating:** 7
**Confidence:** 3

**Summary:**

The manuscript studies the challenges caused by out-of-distribution (OOD) data in machine learning models. They identify "the difference between the predicted and true OOD label distributions" (called pseudo-label shift) as the main reason many existing works underestimate OOD errors. They propose a new approach called Confidence Optimal Transport (COT), which uses optimal transport theory to provide more robust error estimates in the presence of pseudo-label shift. Another approach they propose is COT with Thresholding (COTT), which further applies thresholding to individual transport costs to improve the accuracy of COT's error estimates. The authors present impressive experiments to conclude that the COTT approach outperforms existing state-of-the-art OOD error-predicting methods. Predicting a classifier's performance on out-of-distribution (OOD) data is a significant challenge in machine learning, so this manuscript is timely and important.

**Strengths:**

- Due to the use of the Wassienstien distance, COT is robust to miscalibration, meaning it performs well even when the predicted probabilities of the model are not well-calibrated. This is a significant advantage as most other metrics are sensitive to miscalibration.

- It looks like COT and COTT are better at handling pseudo-label shifts, i.e., when there are differences between the predicted and true OOD label distributions. Table 1 extensively compares COT and COTT against other metrics for measuring the OOD errors.

- It is very welcomed that COT provides some performance guarantee. Under the assumption that the true and predicted label distributions are the same, COT predicts an error of less than half the Wasserstein distance between the pseudo-label distribution and the true label distribution

**Weaknesses:**

- The computation of COT and COTT involves solving a linear program of optimal transport. Yes, the authors propose a batching technique to make their methods scalable, but there is a concern that COTT is computationally infeasible for large-scale inputs.

- In most practical setups, COT doesn't have any performance guarantees because the assumption that the true and predicted label distributions are the same is rarely valid.

**Questions:**

- The performance of COTT depends on the threshold. The choice of the threshold could impact the performance of COTT. How sensitive is COTT to the threshold value? Is thresholding effective in reducing the impact of outliers?

**Limitations:**

The paper has a particular section addressing the limitations of this work.

---

> ### Author Rebuttal · Authors · 2023-08-09
>
> We thank the reviewer for the thoughtful review and for the recognition that our methods “have a significant advantage” over other metrics, our experiments are extensive, and we provide performance guarantees. We would like to address the concerns and questions raised by the reviewer as follows:
>
> ### On Weakness 1
>
> We thank the reviewer for raising concerns over the scalability of our algorithm. Indeed, using COT/COTT requires solving a linear program with $O(n^3)$ time complexity. That’s why we proposed the batched version of COT and conducted all experiments with this setup. For a batch of 10,000, COT finishes the prediction in 7 seconds when using the Python Optimal Transport (POT) library [1].  Thus, we think batched COT is reasonably fast, especially considering the current problem setting where we want to estimate the performance of a bunch of samples instead of single ones, which, most likely, do not require real-time response. We have added this discussion to our paper.
>
> ### On Weakness 2
>
> We assume the reviewer refers to our assumption that the target label marginal is similar to the source label marginal. In that case, we agree that this assumption may not hold in real-world cases. That’s why we conducted experiments where we simulate mild label shift in Section B4 of our supplemental material and demonstrated that COT/COTT still provides superior performance. We also agree that COT/COTT might fail in extreme cases but ultimately, all works tackling distribution shift have to make some assumptions about the data as it is theoretically impossible to have an OOD error prediction algorithm that works for any distribution shift (See section 3.1 in Garg et al. 2022 [2] ). Thus, we think one significance of our work is that we specify a specific assumption that, when held true, makes our method a more desirable one compared to other methods.
>
> ### On Question 1
>
> We thank the reviewer for this great question regarding the threshold of COTT. We would like to point out that in our implementation, the threshold of COTT is not a hyperparameter, but is learned based on the transportation costs such that the number of validation samples whose costs are above the threshold matches the number of mistakes (See Section 3.2, paragraph 3 of our paper). While it is interesting to see the impact of varying the threshold, the current way of setting the threshold makes it a consistent estimator of in-distribution test errors (see Supp D.3 in Garg et al. 2022 [2]).  Empirically, thresholding is effective in reducing the impact of outliers as evidenced by the performance improvement of COTT over COT, and ATC over AC.
>
> ### References
>
> [1] Flamary, Rémi, et al. "Pot: Python optimal transport." The Journal of Machine Learning Research 22.1 (2021): 3571-3578.
>
> [2] Garg, S. and Balakrishnan, S., 2022. Leveraging Unlabeled Data to Predict Out-of-Distribution Performance. ICLR.

---

> > ### Comment · Reviewer_HB65 · 2023-08-13
> > **Thanks for the response**
> >
> > Dear Authors, thanks a lot for the detailed responses. I decide the keep my scorings.

---

### Author Rebuttal · Authors · 2023-08-10

We would like to thank all reviewers for their time and effort in reviewing our work. We are glad that our work receives the following compliments from our reviewers:
- It shows “clear advantage of COT and COTT for predicting OOD error” (1Jqy) over “most other metrics that are sensitive to miscalibration” (HB65), and “provides some performance guarantee” (HB65).
- “The motivation in this work is clear” (chpC), “addresses an important problem and provides a highly innovative perspective” (dn4Z), “the idea of using pseudo-label shift to characterize error is natural and straight-forward” (1ruP), and “the proposed method is not very complicated” (1Jqy), "this manuscript is timely and important" (HB65).
- “The authors present impressive experiments” (HB65), “experiments results are reasonable” (1ruP), “experiments are compelling and make a strong point” (dn4Z),  “the code is available which improves the reproducibility” (chpC).

We have responded to all reviewer concerns and completed all the additional experiments that were requested. The attached PDF file contains experimental results for ColoredMNIST, per reviewer 1Jqy’s request.

---

### Decision · Program_Chairs · 2023-09-21

**Decision:**

Accept (poster)

**Comment:**

This paper offers a fresh perspective on addressing the OOD problem, specifically focusing on the pseudo-label shift. It presents a rigorous theoretical analysis accompanied by valid empirical results, garnering approval from all reviewers. However, some statements could benefit from further clarification for enhanced presentation. Overall, the AC is inclined to recommend acceptance.